# Domain Adaptation for Large-Vocabulary Object Detectors

**Kai Jiang**[1,†]**, Jiaxing Huang**[2,†]**, Weiying Xie**[1]**, Jie Lei**[3]**, Yunsong Li**[1]**, Ling Shao**[4]**, Shijian Lu**[2,*]
[1]State Key Laboratory of Integrated Services Networks, Xidian University, Xi'an 710071, China
[2]S-lab, School of Computer Science and Engineering, Nanyang Technological University
[3]School of Electrical and Data Engineering at the University of Technology Sydney
[4]UCAS-Terminus AI Lab, University of Chinese Academy of Sciences, China

## Abstract

Large-vocabulary object detectors (LVDs) aim to detect objects of many categories, which learn super objectness features and can locate objects accurately while applied to various downstream data. However, LVDs often struggle in recognizing the located objects due to domain discrepancy in data distribution and object vocabulary. At the other end, recent vision-language foundation models such as CLIP demonstrate superior open-vocabulary recognition capability. This paper presents KGD, a Knowledge Graph Distillation technique that exploits the implicit knowledge graphs (KG) in CLIP for effectively adapting LVDs to various downstream domains. KGD consists of two consecutive stages: 1) KG extraction that employs CLIP to encode downstream domain data as nodes and their feature distances as edges, constructing KG that inherits the rich semantic relations in CLIP explicitly; and 2) KG encapsulation that transfers the extracted KG into LVDs to enable accurate cross-domain object classification. In addition, KGD can extract both visual and textual KG independently, providing complementary vision and language knowledge for object localization and object classification in detection tasks over various downstream domains. Experiments over multiple widely adopted detection benchmarks show that KGD outperforms the state-of-the-art consistently by large margins.

## 1 Introduction

Object detection aims to locate and classify objects in images, which conveys critical information about "what and where objects are" in scenes. It is very important in various visual perception tasks in autonomous driving, visual surveillance, object tracking, etc. Unlike traditional object detection, large-vocabulary object detection [1, 2, 3] aims to detect objects of a much larger number of categories, e.g., 20k object categories in [3]. It has achieved very impressive progress recently thanks to the availability of large-scale training data. On the other hand, large-vocabulary object detectors (LVDs) often struggle while applied to various downstream tasks as their training data often have different distributions and vocabularies as compared with the downstream data, i.e., due to domain discrepancies.

In this work, we study unsupervised domain adaptation of LVDs, i.e., how to adapt LVDs towards various downstream tasks with abundant unlabelled data available. Specifically, we observe that LVDs learn superb generalizable objectness knowledge from massive object boxes, being able to locate objects in various downstream images accurately [3]. However, LVDs often fail to classify

---

[†] These authors contributed equally to this work.
[*] Corresponding author.

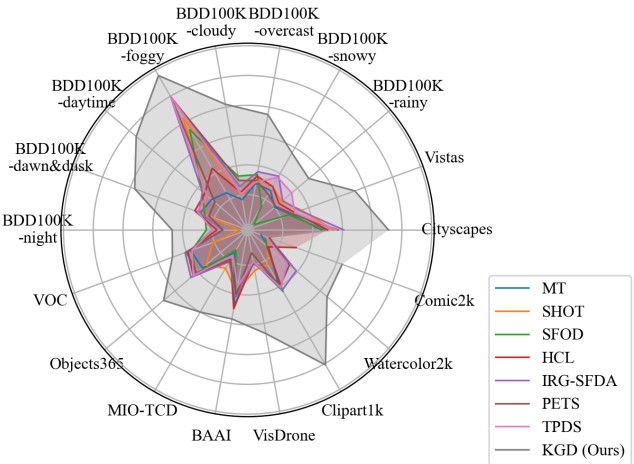

Figure 1: A comparison of the domain adaptation performance of our method against existing methods. Our method outperforms the state-of-the-art consistently on 11 widely studied downstream detection datasets in terms of AP50 improvements. The results of all methods are acquired with the same baseline [3].

the located object due to two major factors: 1) the classic dataset-specific class-imbalance and the resultant distribution bias across domains; and 2) different vocabularies across domains [4, 5]. At the other end, vision-language models (VLMs) [6] such as CLIP [7] learn from web-scale images and text of arbitrary categories, which achieve significant generalization performance in various downstream tasks with severe domain shifts. Hence, effective adaptation of LVDs towards various unlabelled downstream domains could be facilitated by combining the superior object localization capability from LVDs and the super-rich object classification knowledge from CLIP.

We design Knowledge Graph Distillation (KGD) that explicitly retrieves the classification knowledge of CLIP to adapt LVDs while handling various unlabelled downstream domains. KGD works with one underling hypothesis, i.e., the generalizable classification ability of CLIP largely comes from its comprehensive knowledge graph learnt over billions of image-text pairs, which enables it to classify objects of various categories accurately. In addition, the knowledge graph in CLIP is implicitly encoded in its learnt parameters which can be exploited in two steps: 1) Knowledge Graph Extraction (KGExtract) that employs CLIP to encode downstream data as nodes and computes their feature distances as edges, constructing an explicit CLIP knowledge graph that captures inherent semantic relations as learnt from web-scale image-text pairs; and 2) Knowledge Graph Encapsulation (KGEncap) that encapsulates the extracted knowledge graph into object detectors to enable accurate object classification by leveraging relevant nodes in the CLIP knowledge graph.

The proposed KGD allow multi-modal knowledge distillation including Language Knowledge Graph Distillation (KGD-L) and Vision Knowledge Graph Distillation (KDG-V). Specifically, KGD-L considers texts as nodes and the distances among text embeddings as edges, enabling detectors to reason whether a visual object matches a text by leveraging other relevant text nodes. KGD-V takes a category of images as a node and the distances among image embeddings as edges, which enhances detection by conditioning on other related visual nodes. Hence, KGD-L and KGD-V complement each other by providing orthogonal knowledge from language and vision perspectives. In this way, KGD allows to explicitly distill generalizable knowledge from CLIP to facilitate unsupervised adaptation of large-vocabulary object detectors towards distinctive downstream datasets.

In summary, the major contributions of this work are threefold. *First*, we propose a knowledge transfer framework that exploits CLIP for effective adaptation of large-vocabulary object detectors towards various unlabelled downstream data. To the best of our knowledge, this is the first work that studies distilling CLIP knowledge graphs for the object detection task. *Second*, we design novel knowledge graph distillation techniques that extracts visual and textual knowledge graphs from CLIP and encapsulates them into object detection networks successfully. *Third*, extensive experiments show

that KGD outperforms the state-of-the-art consistently across 11 widely studied detection datasets as shown in Fig. 1.

## 2 Related works

**Large-vocabulary Object Detection** [8, 9, 10, 11, 12, 13, 14] aims to detect objects of thousands of classes. Most existing studies tackle this challenge by designing various class-balanced loss functions [8] for effective learning from large-vocabulary training data and handling the long-tail distribution problem [15, 16, 17, 18]. Specifically, several losses have been proposed, such as Equalization losses [19, 20], SeeSaw loss [21], and Federated loss [22]. On the other hand, [23] and Detic [3] attempt to introduce additional image-level datasets with large-scale fine-grained classes for training large-vocabulary object detector (LVD), aiming to expand the detector vocabulary to tens of thousands of categories. These LVDs learn superb generalizable objectness knowledge from object boxes of massive categories and are able to locate objects in various downstream images accurately [3]. However, they often fail to classify the located objects [4, 5] accurately. In this work, we focus on adapting LVDs towards various unlabelled downstream data by utilizing the super-rich object classification knowledge from CLIP.

**Domain Adaptation** aims to adapt source-trained models towards various target domains. Previous work largely focuses on unsupervised domain adaptation (UDA), which minimizes the domain discrepancy by discrepancy minimization [24, 25], adversarial training [26, 24, 27, 28, 29, 30], self-supervised learning [31, 32, 33, 34], or self-training [35, 36, 37, 38, 39, 40, 41, 42, 43]. Recently, source-free domain adaptation (SFDA) generates pseudo labels for target data without accessing source data, which performs domain adaptation with entropy minimization [44], self training [45, 46, 47, 48, 49], contrastive learning [50, 51, 52, 53], etc. However, most existing domain adaptation methods struggle while adapting LVDs toward downstream domains, largely due to the low-quality pseudo labels resulting from the discrepancy in both data distributions and object vocabulary.

**Vision-Language Models (VLMs)** have achieved great success in various vision tasks [6]. They are usually pretrained on web-crawled text-image pairs with a contrastive learning objective. Representative methods such as CLIP [7] and ALIGN [54] have demonstrated very impressive generalization performance in many downstream vision tasks. Following [7, 54], several studies [54, 55, 56, 57] incorporate cross-attention layers and self-supervised objectives for better cross-modality modelling of noisy data. In addition, several studies [58, 59, 60, 61] learn fine-grained and structural alignment and relations between image and text. In this work, we aim to leverage the generalizable knowledge learnt by VLMs to help adapt LVDs while handling various unlabelled downstream data.

**Knowledge Graph (KG)** [62] is a semantic network that considers real-world entities or concepts as nodes and treats the semantic relations among them as edges. Multi-modal knowledge graph [63, 64] extends knowledge from text to the visual domain, enhancing machines' ability to describe and comprehend the real world. These KGs have proven great effectiveness in storing and representing factual knowledge, leading to successful applications in various fields such as entity recognition [65, 66], question-answering [67], and information retrieval [68]. Different from the aforementioned KGs and MMKGs that are often handcrafted by domain experts, we design knowledge graph distillation that builds a LKG and a VKG by explicitly retrieving VLM's generalizable knowledge learnt from web-scale image-text pairs, which effectively uncover the semantic relations across various textual and visual concepts in different downstream tasks, ultimately benefiting the adaptation of LVDs.

## 3 Method

**Task Definition.** This paper focuses on unsupervised adaptation of large-vocabulary object detectors (LVDs). We are provided with a set of unlabeled downstream domain data $\mathcal{D}_t = \{\mathbf{x}_i^t\}_{i=1}^{N_t}$ and an LVD pre-trained on labeled source domain detection dataset $\mathcal{D}_s = \{\mathbf{x}_i^s, \mathbf{y}_i^s\}_{i=1}^{N_s}$. $\mathbf{x}_i$ and $\mathbf{y}_i = \{(\mathbf{p}_j, \mathbf{t}_j)\}_{j=1}^{M}$ are the image and $M$ instance annotations of $i$-th sample, where $\mathbf{p}_j$ and $\mathbf{t}_j$ denote the ground-truth category and box coordinate of $j$-th instance. $N_s$ and $N_t$ refer to the number of samples in $\mathcal{D}_s$ and $\mathcal{D}_t$. The goal is to adapt the pretrained LVD towards the downstream domain $\mathcal{D}_t$ by using the unlabelled images.

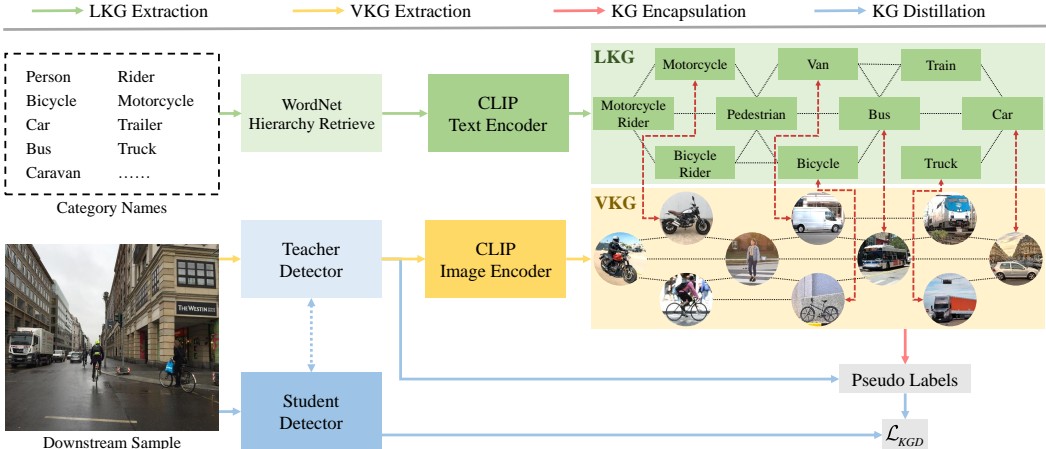

Figure 2: Overview of the proposed Knowledge Graph Distillation (KGD). KGD comprises two consecutive stages including Knowledge Graph Extraction (KGExtract) and Knowledge Graph Encapsulation (KGEncap). KGExtract employs CLIP to encode downstream data as nodes and considers their feature distances as edges, explicitly constructing KGs including language knowledge graph (LKG) and vision knowledge graph (VKG) that inherit the rich semantic relations in CLIP. The dashed reddish lines between LKG and VKG represent the cross-modal edges that connect the nodes between vision and language modalities, enabling the integration of both language and visual information. KGEncap transfers the extracted KGs into the large-vocabulary object detector to enable accurate object classification over downstream data. Besides, KGD works for both image and text data and allow extracting and transferring vision KG (VKG) and language KG (LKG), providing complementary knowledge for adapting large-vocabulary object detectors for handling various unlabelled downstream domains.

**Naïve Solution with Mean Teacher Method (MT) [45].** In this paper, we adopt Detic [3] as the pretrained LVD, which utilizes CLIP text embeddings as the classifier. We employ mean teacher [45] as the preliminary solution, which involves a teacher detector and a student detector where the former generates pseudo labels to train the latter while the latter updates the former in a momentum manner. Given a batch of $B$ unlabeled target samples, the teacher detector $\Phi_t$ first produces detection predictions on them, which are then filtered with a predefined threshold $\tau$ to generate detection pseudo label $\hat{\mathbf{y}}_i$ (consisting of classes and bounding boxes). With $\hat{\mathbf{y}}_i$, the unsupervised training of student detector $\Phi_s$ on the unlabeled downstream data can be formulated as the following:

$$Loss = \frac{1}{B} \sum_{i=1}^{B} \mathcal{L}\left(\Phi_s(\mathbf{x}_i^t), \hat{\mathbf{y}}_i\right), \tag{1}$$

where $\mathcal{L}(\cdot) = \mathcal{L}_{rpn}(\cdot) + \mathcal{L}_{reg}(\cdot) + \mathcal{L}_{cls}(\cdot)$ is the detection loss function in which $\mathcal{L}_{rpn}(\cdot)$, $\mathcal{L}_{reg}(\cdot)$, and $\mathcal{L}_{cls}(\cdot)$ denote the loss for region proposal network, regression, and classification, respectively. Note both teacher detector $\Phi_t$ and student detector $\Phi_s$ are initialized with the pretrained LVD.

**Motivation.** On the other hand, although the LVD is able to locate objects in various downstream-domain images accurately [3], it often fails to classify the located objects, leading to very noisy detection pseudo labels when serving as the teacher detector. At the other end, vision-language models (VLMs) [6] such as CLIP [7] learns from web-scale images-text pairs of arbitrary categories, which possesses the ability to classify objects accurately in various downstream data. Thus, we argue that effective adaptation of LVDs towards various unlabelled downstream data could be facilitated by combining the superior object localization capability from LVDs and the super-rich object classification knowledge from CLIP. To this end, we design Knowledge Graph Distillation (KGD) with Language KGD and Vision KGD, aiming to explicitly retrieves the classification knowledge of CLIP to adapt LVDs while handling various unlabelled downstream data. The overview of our proposed KGD is shown in Fig. 2.

### 3.1 Language knowledge graph distillation

The proposed language knowledge graph distillation (KGD-L) aims on distilling knowledge graph from the perspective of text modality. KGD-L works in a two-step manner. The first step is language knowledge graph (LKG) extraction with a large lexical database named WordNet [69] that aims to uncover the implicitly encoded language knowledge in CLIP. With the guidance from the WordNet that stores a wide range of knowledge, LKG Extraction builds a category-discriminative and domain-generalizable LKG. The second step is LKG encapsulation that encapsulates the extracted LKG into the teacher detector, enabling the detector to reason whether a visual object matches a text by leveraging other relevant text nodes and ultimately generate more accurate detection pseudo labels.

**LKG Extraction with WordNet Hierarchy.** We first generate domain-generalizable [70] prompts for each object category by leveraging the large lexical database WordNet [69]. Specifically, given the category set $\mathcal{C} = \{\mathbf{c}_i | i = 1 \dots, N_c\}$ of a downstream domain, we obtain the WordNet [69] Synset definition as well as the hyponym set of category $\mathbf{c}_i$ as follows:

$$\mathbf{d}_i, \mathcal{S}_i = \text{WNRetrieve}(\mathbf{c}_i), \tag{2}$$

where $\text{WNRetrieve}(\cdot)$ retrieves the WordNet database [69] and returns the definition $\mathbf{d}_i$ as well as the hyponym set $\mathcal{S}_i$ of its input. $\mathcal{S}_i = \{\mathbf{s}_j\}_{j=1}^m$, where $\mathbf{s}_j$ refers to the $j$th hyponym of category $\mathbf{c}_i$ and $m$ refers to the cardinal number of $\mathcal{S}_i$. Note that hyponym $\mathbf{s}_j$ is the concatenation of the class name and its descriptions. In this way, a category name $\mathbf{c}_i$ can be better defined and described with the informative yet accurate category definition in its hyponym set from WordNet, which are then combined with $\mathbf{d}_i$ as a set of domain generalizable prompts for category $\mathbf{c}_i$:

$$\tilde{\mathcal{S}}_i = \mathcal{S}_i \cup \{\mathbf{d}_i\}, \tag{3}$$

and the domain generalizable prompt set of category set $\mathcal{C}$ can be constructed as the following:

$$\tilde{\mathcal{S}} = \bigcup_{i=1}^{N_c} \tilde{\mathcal{S}}_i. \tag{4}$$

With the category-discriminative and domain-generalizable information contained in $\tilde{\mathcal{S}}$, we formulate the proposed LKG as a weighted undirected graph $G_L = (V_L, U_L, \Omega)$, which is capable of capturing semantic relationships and associations between different category concepts. $V_L = \{\tilde{\mathbf{s}}_i\}_{i=1}^{N_c(m+1)}$ is the vertex set in which each node $\tilde{\mathbf{s}}_i$ refers to a description in $\tilde{\mathcal{S}}$. And $U_L = \{(\tilde{\mathbf{s}}_i, \tilde{\mathbf{s}}_j)\}$ is the edge set. $\Omega$ is a matrix of node feature vectors $\Omega_i = T(\tilde{\mathbf{s}}_i)$, where $T(\cdot)$ denotes the CLIP text encoder.

**LKG Encapsulation** encapsulates the comprehensive knowledge in the extracted LKG into the teacher detector to facilitate detection pseudo label generation. Specifically, we first employ CLIP to encode the regions cropped by the teacher detector and then generate pseudo labels for each region feature conditioned on LKG. Given the image $\mathbf{x}^t \in \mathcal{D}_t$, we feed it into the teacher detector $\Phi_t$ to acquire the prediction as the following:

$$\hat{\mathbf{y}} = \Phi_t(\mathbf{x}^t), \tag{5}$$

where $\hat{\mathbf{y}} = \{(\hat{\mathbf{p}}_j, \hat{\mathbf{t}}_j)\}_{j=1}^M$, $\hat{\mathbf{p}}_j$ denotes the probability vector of the predicted bounding box $\hat{\mathbf{t}}_j$ after Softmax activation function. $M$ denotes the number of predicted proposals after the thresholding with $\tau$, i.e., a predicted proposal will be discarded if its confidence score is less than $\tau$.

Next, we employ CLIP to encode the predicted object proposals in $\hat{\mathbf{y}}$ as follows:

$$F = V\left(Crop\left(\mathbf{x}^t, \hat{\mathbf{y}}\right)\right), \tag{6}$$

where $Crop(\cdot)$ crops square regions from image $\mathbf{x}^t$ based on the longer edges of bounding boxes in $\hat{\mathbf{y}}$, $V(\cdot)$ is the image encoder of CLIP, and the $j$-th vector $\mathbf{f}_j$ of matrix $F$ is the feature of $j$-th proposal in $\hat{\mathbf{y}}$.

With the extracted LKG $G_L$ and the features of objects (or object proposals) $F$, we reason the class of objects conditioned on $G_L$ with a two-layer graph convolutional network (GCN) [71] as follows:

$$[Q^F; Q^\Omega] = \text{Softmax}(D^{-\frac{1}{2}} A D^{-\frac{1}{2}} \text{ReLU}(D^{-\frac{1}{2}} A D^{-\frac{1}{2}} H^0 W^0) W^1), \tag{7}$$

where $H^0 = [F; \Omega]$, $A_{ij} = exp(-||H_i^0 - H_j^0||_2^2 / \text{Var}(||H_i^0 - H_j^0||_2^2))$, $A_{ii} = 1$, and $D_{ii} = \sum_j A_{ij}$. $Q_{ji}^F / Q_{ji}^\Omega$ is the $i$-th element in probability vector $Q_j^F / Q_j^\Omega$, which denotes the predicted category

probability of being $\mathbf{c}_i$ for object feature $\mathbf{f}_j$/LKG node $\tilde{\mathbf{s}}_j$. $\{W^l\}_{l=0}^1$ are the trainable weights. For updating $\{W^l\}_{l=0}^1$, we minimizing the following cross entropy error over the nodes in LKG:

$$\mathcal{L}_{LKG}(\mathbf{x}^t) = -\sum_i \sum_j \left( log(Q_{ji}^\Omega) \cdot \mathbb{I}(\tilde{\mathbf{s}}_j \in \tilde{S}_i) \right). \tag{8}$$

Then we encapsulate the extracted LKG into $\Phi_t$ by,

$$\mathbf{p}_{ji}^l = \hat{\mathbf{p}}_{ji} \cdot Q_{ij}^F, \tag{9}$$

where $\hat{\mathbf{p}}_{ji}$ is the $i$-th element in probability vector $\hat{\mathbf{p}}_j$, which denotes the predicted category probability of $\mathbf{c}_i$. The first term in Eq. 9 denotes the original prediction probability from the teacher model while the second term in Eq. 9 stands for the prediction probability from LKG. $\mathbf{p}_{ji}^l$ denotes the prediction probability calibrated by LKG.

In this way, KGD-L extracts and encapsulates LKG from CLIP into the teacher detector, enabling it to reason whether an object matches a category conditioned on the relevant nodes in LKG and ultimately refining the original detection pseudo labels.

## 3.2 Vision knowledge graph distillation

As LKG captures language knowledge only, we further design vision knowledge graph distillation (KGD-V) that extracts a vision knowledge graph (VKG) and encapsulates it into the teacher detector to improve pseudo label generation. Specifically, VKG captures vision knowledge dynamically along the training process, which complement LKG by providing orthogonal and update-to-date vision information.

**Dynamic VKG Extraction.** We first initialize VKG with the CLIP text embedding and then employ the update-to-date object features to update it using manifold smoothing. Specifically, we initialize VKG as a weighted undirected graph $G_V = (V_V, U_V)$, in which each node $\mathbf{v}_i \in V_V$ is initialized with the CLIP text embedding of category $\mathbf{c}_i$:

$$\mathbf{v}_i = T(\mathbf{c}_i), \tag{10}$$

and the graph edge $u_{ij} \in U_V$ is defined as the cosine similarity between nodes $\mathbf{v}_i$ and $\mathbf{v}_j$. Given a batch of $\{\mathbf{x}_b^t\}_{b=1}^B \subseteq \mathcal{D}_t$ and the corresponding pseudo labels $\{\hat{\mathbf{y}}_b\}_{b=1}^B$ and CLIP features $\{F_b\}_{b=1}^B$, the visual embedding centroid of category $\mathbf{c}_k$ can be obtained as the following:

$$\boldsymbol{\theta}_i = \frac{\sum_{b=1}^B \sum_{\mathbf{f}_j \in \mathbf{F}_b} \mathbf{f}_j \cdot \mathbb{I}(\hat{\mathbf{p}}_j(i) == \hat{\mathbf{p}}_j^{max})}{\sum_{b=1}^B \sum_{\mathbf{f}_j \in \mathbf{F}_b} \mathbb{I}(\hat{\mathbf{p}}_j(i) == \hat{\mathbf{p}}_j^{max})}, \tag{11}$$

where $\hat{\mathbf{p}}_j^{max}$ is the maximum element in probability vector $\hat{\mathbf{p}}_j$, $\mathbb{I}$ is the indicator function. And an affinity matrix $A$ can be calculated as $A_{ij} = exp(-r_{ij}^2/\sigma^2)$ and $A_{ii} = 0$, where $r_{ij} = ||\boldsymbol{\theta}_i - \boldsymbol{\theta}_j||_2$ and $\sigma^2 = \text{Var}(r_{ij}^2)$. In each iteration, the node of VKG is preliminarily updated as:

$$\mathbf{v}_i \leftarrow \lambda \mathbf{v}_i + (1 - \lambda)\boldsymbol{\theta}_i. \tag{12}$$

In order to incorporate the downstream visual graph knowledge into VKG, we perform additional steps to smooth the node of VKG, using the affinity matrix $A$ from the current batch as a guide:

$$\mathbf{v}_i = \sum_j W_{ij} \mathbf{v}_j, \tag{13}$$

where $W = (I - \alpha L)^{-1}$, $L = D^{-\frac{1}{2}} A D^{-\frac{1}{2}}$, $D_{ii} = \sum_j A_{ij}$, $\alpha$ is a scaling factor set as [72], and $I$ is the identity matrix.

**VKG Encapsulation** encapsulate the orthogonal and update-to-date vision knowledge in the extracted VKG into the teacher detector, which complements LKG and further improves pseudo label generation. With the extracted dynamic VKG $G_V$ and the object features $F$ in image $\mathbf{x}^t$, we encapsulate the extracted VKG into $\Phi_t$ in a similar way as the LKG Encapsulation as follows:

$$\mathbf{p}_{ji}^v = \hat{\mathbf{p}}_{ji} \cdot \frac{exp(cos \langle \mathbf{f}_j, \mathbf{v}_i \rangle)}{\sum_{i'} exp(cos \langle \mathbf{f}_j, \mathbf{v}_{i'} \rangle)}, \tag{14}$$

Table 1: Benchmarking over autonomous driving datasets under various weather and time conditions. † signifies that the methods employ WordNet to retrieve category definitions given category names, and CLIP to predict classification pseudo labels for objects. We adopt AP50 in evaluations. The results of all methods are acquired with the same baseline [3] as shown in the first row.

| Method | Cityscapes [73] | Vistas [74] | BDD100K-weather [75] | | | | | BDD100K-time-of-day [75] | | |
|---|---|---|---|---|---|---|---|---|---|---|
| | | | rainy | snowy | overcast | cloudy | foggy | daytime | dawn&dusk | night |
| Detic [3] (Baseline) | 46.5 | 35.0 | 34.3 | 33.5 | 39.1 | 42.0 | 28.4 | 39.2 | 35.3 | 28.5 |
| MT [45] | 49.1 | 35.7 | 34.3 | 34.2 | 39.9 | 41.7 | 28.9 | 40.0 | 36.3 | 28.5 |
| MT [45]† | 50.0 | 36.6 | 35.0 | 35.3 | 40.9 | 43.0 | 29.8 | 42.1 | 38.4 | 29.1 |
| SHOT [44] | 49.9 | 36.5 | 34.9 | 34.5 | 40.2 | 42.0 | 34.7 | 40.5 | 36.1 | 26.7 |
| SHOT [44]† | 50.8 | 37.4 | 36.1 | 35.7 | 41.8 | 44.1 | 35.6 | 42.4 | 38.1 | 28.0 |
| SFOD [46] | 49.3 | 35.6 | 32.5 | 33.0 | 40.5 | 43.3 | 33.8 | 40.8 | 36.0 | 28.9 |
| SFOD [46]† | 50.3 | 36.6 | 33.6 | 33.8 | 42.8 | 45.6 | 34.7 | 43.4 | 37.9 | 30.1 |
| HCL [50] | 49.5 | 36.0 | 34.7 | 34.5 | 40.4 | 42.2 | 30.8 | 40.6 | 36.7 | 28.2 |
| HCL [50]† | 50.7 | 37.0 | 35.6 | 35.7 | 42.2 | 44.3 | 31.9 | 42.9 | 38.6 | 29.5 |
| IRG-SFDA [51]† | 50.6 | 36.4 | 35.0 | 35.3 | 40.7 | 42.6 | 36.4 | 40.8 | 36.4 | 27.8 |
| IRG-SFDA [51]† | 51.7 | 37.5 | 35.9 | 36.4 | 42.6 | 44.8 | 36.7 | 43.0 | 38.3 | 28.9 |
| PETS [76] | 50.2 | 35.8 | 34.4 | 33.9 | 40.1 | 43.0 | 36.3 | 39.7 | 35.7 | 27.8 |
| PETS [76]† | 50.8 | 37.4 | 35.9 | 36.3 | 41.0 | 42.8 | 36.7 | 40.9 | 37.2 | 27.7 |
| TPDS [77] | 50.1 | 36.0 | 35.8 | 35.2 | 40.0 | 42.1 | 36.4 | 40.4 | 36.5 | 28.5 |
| TPDS [77]† | 50.3 | 37.1 | 35.6 | 35.9 | 40.5 | 43.4 | 36.9 | 41.3 | 36.7 | 28.9 |
| **KGD (Ours)** | **53.6** | **40.3** | **37.3** | **37.1** | **44.6** | **48.2** | **38.0** | **46.6** | **41.0** | **31.2** |

where $\hat{\mathbf{p}}_{ji}$ is the $i$-th element in vector $\hat{\mathbf{p}}_j$, denoting the predicted probability of category $\mathbf{c}_i$. The first term in Eq. 14 is the prediction probability from the teacher model while the second term in Eq. 14 is the prediction probability from VKG. $\mathbf{p}_{ji}^v$ is the prediction probability calibrated by VKG.

In this way, KGD-V extracts and encapsulates the VKG from CLIP into the teacher detector, further refining the detection pseudo labels of visual objects by conditioning on related visual nodes in VKG.

### 3.3 Overall objective

Finally, with the pseudo labels $\mathbf{p}_j^l$ and $\mathbf{p}_j^v$ generated from KGD-L and KGD-V respectively, the unsupervised training loss of KGD can be formulated as the following:

$$\mathcal{L}_{KGD} = \sum_{\mathbf{x}^t \in \mathcal{D}_t} \left( \mathcal{L}\left(\Phi_s(\mathbf{x}^t), \tilde{\mathbf{y}}\right) + \mathcal{L}_{LKG}(\mathbf{x}^t) \right), \quad (15)$$

where $\tilde{\mathbf{y}} = \{(\tilde{\mathbf{p}}_j, \hat{\mathbf{t}}_j)\}_{j=1}^M$, and $\tilde{\mathbf{p}}_j = N(\mathbf{p}_j^l + \mathbf{p}_j^v)$. $N(\cdot)$ normalizes data to range $[0, 1]$. The pseudo code of the proposed KGD is provided in appendix.

## 4 Experiments

This section presents experimental results. Dataset details and implementation details can be found in the Appendix. Section 4.1 presents the experiments across various downstream domain datasets. Section 4.2 and Section 4.3 provide ablation studies and discuss different features of KGD.

### 4.1 Results

Tables 1-2 show the benchmarking of our methods with state-of-the-art domain adaptive detection methods. As there are few prior studies on LVD adaptation, we compare our proposed method with state-of-the-art source-free domain adaptation methods for benchmarking, including Mean Teacher (MT) [45], SHOT [44], SFOD [46], HCL [50], IRG-SFDA [51], PETS [76], and TPDS [77]. For fair comparison, we incorporate CLIP [7] and WordNet [69] into the compared methods (marked with †). Specifically, we employ WordNet [69] to generate category definitions given category names, and CLIP [7] to predict pseudo labels for object classification.

**Object detection for autonomous driving.** As Table 1 shows, the proposed KGD outperforms the baseline substantially over the general autonomous driving datasets Cityscapes and Vistas (with an average improvement of 6.20 in AP50). KGD also outperforms the state-of-the-art by 2.35 on average, demonstrating the superiority of KGD in adapting pretrained LVDs toward autonomous driving scenarios with substantial inter-domain discrepancy. In addition, Table 1 shows experiments on autonomous driving data under various weather and time conditions. We can observe that KGD still achieves superior detection performance even though the unlabeled target data experience large style variation and severe quality degradation. Further, the experiments show that KGD still outperforms

Table 2: Benchmarking over common objects datasets, intelligent surveillance datasets, and artistic illustration datasets. † signifies that the methods employ WordNet to retrieved category definitions given category names, and CLIP to predict classification pseudo labels for objects. We adopt AP50 in evaluations. The results of all methods are acquired with the same baseline [3] as shown in first row.

| Method | Common Objects | | Intelligent Surveillance | | | Artistic Illustration | | |
|---|---|---|---|---|---|---|---|---|
| | VOC [78] | Objects365 [79] | MIO-TCD[80] | BAAI [81] | VisDrone [82] | Clipart1k [83] | Watercolor2k [83] | Comic2k [83] |
| Detic [3] (Baseline) | 83.9 | 29.4 | 20.6 | 20.6 | 19.0 | 61.0 | 58.9 | 51.2 |
| MT [45] | 85.6 | 31.0 | 20.0 | 23.4 | 18.9 | 62.7 | 58.4 | 49.8 |
| MT [45]† | 86.2 | 31.4 | 20.9 | 23.9 | 20.4 | 63.4 | 59.6 | 51.1 |
| SHOT [44] | 84.0 | 30.7 | 21.2 | 22.5 | 19.4 | 61.3 | 58.3 | 50.4 |
| SHOT [44]† | 84.5 | 31.2 | 22.3 | 23.3 | 20.9 | 62.3 | 59.8 | 52.1 |
| SFOD [46] | 85.5 | 31.6 | 19.8 | 22.8 | 18.8 | 63.4 | 58.2 | 50.1 |
| SFOD [46]† | 86.2 | 32.0 | 21.0 | 23.1 | 20.2 | 64.6 | 59.3 | 51.8 |
| HCL [50] | 85.8 | 31.8 | 20.5 | 23.6 | 18.8 | 63.1 | 58.3 | 52.3 |
| HCL [50]† | 86.5 | 32.3 | 21.1 | 24.1 | 19.6 | 64.7 | 59.7 | 53.7 |
| IRG-SFDA [51] | 86.0 | 32.0 | 20.7 | 22.8 | 18.8 | 63.3 | 60.8 | 50.4 |
| IRG-SFDA [51]† | 86.3 | 32.3 | 21.6 | 23.7 | 20.0 | 65.0 | 61.5 | 52.0 |
| PETS [76] | 85.9 | 31.5 | 20.6 | 22.6 | 18.2 | 63.0 | 60.2 | 50.4 |
| PETS [76]† | 86.3 | 32.1 | 21.1 | 23.2 | 19.3 | 63.6 | 61.3 | 50.6 |
| TPDS [77] | 85.5 | 31.8 | 20.2 | 22.1 | 18.8 | 63.1 | 60.0 | 50.1 |
| TPDS [77]† | 85.6 | 32.0 | 21.1 | 23.2 | 19.2 | 64.3 | 61.4 | 50.6 |
| **KGD (Ours)** | **86.9** | **34.4** | **24.6** | **24.3** | **23.7** | **69.1** | **63.5** | **55.6** |

Table 3: Ablation studies of KGD with Language Knowledge Graph Distillation (KGD-L) and Vision Knowledge Graph Distillation (KGD-V). The experiments are conducted on the Cityscapes.

| Method | Detic (Baseline) | | KGD (Ours) | |
|---|---|---|---|---|
| Language Knowledge Graph Distillation | | ✓ | | ✓ |
| Vision Knowledge Graph Distillation | | | ✓ | ✓ |
| AP50 | 46.5 | 52.8 | 52.7 | **53.6** |

the state-of-the-art clearly when CLIP and WordNet are incorporated, validating that the performance gain largely comes from our novel KGD instead of merely using CLIP and WordNet.

**Object detection for intelligent surveillance.** The detection results on intelligent surveillance datasets are presented in Table 2. Notably, the proposed KGD surpasses all other methods by significant margins, which underscores the effectiveness of KGD in adapting the pretrained LVD towards the challenging surveillance scenarios with considerable variations in camera lenses and angles. The performance improvements achieved by KGD in this context demonstrate its effectiveness in exploring the unlabeled surveillance datasets by retrieving the classification knowledge of CLIP.

**Object detection for common objects.** We evaluate the effectiveness of our KGD on the common object detection task using Pascal VOC and Objects365. Table 2 reports the detection results, showcasing significant improvements over the baseline and outperforming state-of-the-arts, thereby highlighting the superiority of KGD. Besides, we can observe that the performance improvements on the Pascal VOC dataset and Objects365 dataset are not as significant as those in autonomous driving. This discrepancy is attributed to the relatively smaller domain gap between common objects and the pretraining dataset of LVD.

**Object detection for artistic illustration.** Table 2 reports the detection results on artistic illustration datasets. The proposed KGD outperforms all other methods by substantial margins, which highlights the effectiveness of KGD in adapting the pretrained large-vocabulary object detector towards artistic images that exhibit distinct domain gaps with natural images.

## 4.2 Ablation studies

In Table 3, we conducted ablation studies to assess the individual contribution of our proposed KGD-L and KGD-V on the task of LVD adaptation. The pretrained LVD (i.e., Detic [3] without adaptation) does not perform well due to the significant variations between its pre-training data and the downstream data, As a comparison, either KGD-L or KGD-V brings significant performance

Table 4: Comparisons with existing CLIP knowledge distillation methods on LVD adaptation. For a fair comparison, we incorporate them with Mean Teacher Method (the columns with 'MT+'). The results of all methods are acquired with the same baseline [3] as shown in the first column.

| Method | Detic [3] (Baseline) | MT [45] | MT [45]+VILD [84] | MT [45]+RegionKD [85] | MT [45]+OADP [86] | **KGD (Ours)** |
|---|---|---|---|---|---|---|
| AP50 | 46.5 | 49.1 | 50.6 | 50.2 | 50.2 | **53.6** |

Table 5: Study of different KGD-L strategies. The experiments are conducted on the Cityscapes.

| Method | Detic (Source only) | KGD-L only | | |
|---|---|---|---|---|
| LKG Extraction with category names | | ✓ | | |
| LKG Extraction with WordNet Synset definitions | | | ✓ | |
| LKG Extraction with WordNet Hierarchy | | | | ✓ |
| AP50 | 46.5 | 51.9 | 52.0 | **52.8** |

Table 6: Study of different KGD-L strategies. The experiments are conducted on the Cityscapes.

| Method | Detic (Source only) | KGD-L Only | |
|---|---|---|---|
| LKG Encapsulation by Feature Distance | | ✓ | |
| LKG Encapsulation | | | ✓ |
| AP50 | 46.5 | 49.6 | **52.8** |

improvements (i.e., +6.3 of AP50 and +6.2 of AP50 over the baseline), demonstrating both language and vision knowledge graphs built from CLIP can clearly facilitate the unsupervised adaptation of large-vocabulary object detectors. The combination of KGD-L and KGD-V performs the best clearly, showing that our KGD-L and KGD-V are complementary by providing orthogonal language and vision knowledge for regularizing the unsupervised adaptation of LVDs.

## 4.3 Discussion

**Language knowledge graph (LKG) Extraction strategies.** Our proposed KGD-L introduces the WordNet [69] to uncover the implicitly encoded language knowledge in CLIP [7] and accordingly enables to build a category-discriminative and domain-generalizable Language Knowledge Graph (LKG) as described in Section 3.1. We examine the superiority of the proposed LKG Extraction with WordNet Hierarchy by comparing it with "LKG Extraction with category names" and "LKG Extraction with WordNet [69] Synset definitions", the former builds LKG directly with the category names from downstream datasets while the latter directly builds LKG using WordNet Synset definitions that are retrieved from the WordNet database with category names from downstream datasets. As Table 5 shows, both strategies achieve sub-optimal performance. For "LKG Extraction with category names", the category names are often ambiguous and less informative which degrades adaptation. For "LKG Extraction with WordNet Synset definitions", the used WordNet Synset definitions are more category-discriminative but they often have knowledge gaps with downstream data, limiting adaptation of the pretrained LVDs. As a comparison, our proposed LKG Extraction with WordNet Hierarchy performs clearly better due to the guidance of Synset definitions as well as their hyponym sets that captures more comprehensive structural knowledge from the WordNet hierarchy which helps generate category-discriminative and domain-generalizable LKG and facilitates the adaption of LVDs towards downstream data effectively.

**Language knowledge graph (LKG) Encapsulation strategies.** Our proposed KGD-L encapsulates the comprehensive knowledge in the extracted LKG into the teacher detector to facilitate detection pseudo label generation as described in Section 3.1. We examine the superiority of the proposed LKG Encapsulation by comparing it with "LKG Encapsulation by Feature Distance", which directly calculate and normalize the feature distance between object proposal feature and LKG nodes, and calibrates the original prediction probability from the teacher model using the normalized feature distance. As Table 6 shows, "LKG Encapsulation by Feature Distance" does not perform well in model adaptation, largely because it cannot effectively aggregate and capture semantic relationships and associations between different nodes in our extracted LKG. As a comparison, our proposed LKG Encapsulation shows clear improvements as the language information is adaptively aggregated along the training process stabilizes and improves the model adaptation, validating the performance gain largely comes from our novel LKG Encapsulation designs instead of merely using WordNet [69] embedding.

**Vision knowledge graph distillation (KGD-V) strategies.** Our proposed KGD-V captures the Dynamic vision knowledge graph (VKG) along the training as described in Section 3.2, which complements LKG by providing orthogonal and update-to-date vision information. We examine the proposed Dynamic VKG Extraction by comparing it with "Static VKG Extraction" and "Dynamic VKG Extraction without Smoothing". The former builds a static VKG with CLIP features of image

crops of objects that are predicted by the pretrained LVD before adaptation and it remains unchanged during the LVD adaptation process, while the latter updates the VKG with Eq. (12) but without smoothing (Eq. 13). As Table 7 shows, "Static VKG Extraction" does not perform well in model adaptation, largely because the extracted static VKG is biased towards the pretraining datasets of the LVD and impedes domain-specific adaptation. For "Dynamic VKG Extraction without Smoothing", the nodes in VKG are updated with unlabeled downstream data in Eq. (12), but the downstream visual graph knowledge is not effectively incorporated into VKG, which limits the adaptation of the pretrained LVD. As a comparison, our proposed Dynamic VKG Extraction shows clear improvements as the update-to-date vision information extracted along the training process dynamically stabilizes and improves the model adaptation.

Table 7: Studies of different KGD-V strategies. The experiments are conducted on the Cityscapes.

| Method | Detic (Source only) | KGD-V only | | |
|---|---|---|---|---|
| Static VKG Extraction | | ✓ | | |
| Dynamic VKG Extraction without Smoothing | | | ✓ | |
| Dynamic VKG Extraction | | | | ✓ |
| AP50 | 46.5 | 51.9 | 52.2 | **52.7** |

**Comparisons with existing CLIP knowledge distillation methods for detection.** We compared our KGD with existing CLIP knowledge distillation methods designed for detection tasks. Most existing methods achieve CLIP knowledge distillation by mimicking its feature space, such as VILD [84], RegionKD [85], and OADP [86]. Table 4 reports the experimental results over the Cityscapes dataset, which shows existing CLIP knowledge distillation methods do not perform well in adapting LVDs to downstream tasks. The main reason is that they merely align the feature space between LVDs and CLIP without considering the inherent semantic relationships between different object categories. KGD also performs knowledge distillation but works for LVDs adaption effectively, largely because it works by extracting and encapsulating knowledge CLIP knowledge graphs which enables accurate object classification by leveraging relevant nodes in the knowledge graphs.

Table 8: Parameter analysis of KGD for the pseudo label generation threshold $\tau$.

| $\tau$ | 0.15 | 0.2 | 0.25 | 0.3 | 0.35 |
|---|---|---|---|---|---|
| AP50 | 53.4 | 53.2 | 53.6 | 53.9 | 53.5 |

**Parameter studies.** In the pseudo label generation in KGD, the reliable pseudo labels are acquired with a pre-defined confidence threshold $\tau$. We studied $\tau$ by changing it from 0.15 to 0.35 with a step of 0.05. Table 8 reports the experiments over the Cityscapes dataset. It shows that $\tau$ does not affect KGD clearly, demonstrating the proposed KGD is tolerant to hyper-parameters.

**Qualitative experimental results.** We present qualitative results of KGD over diverse downstream domain detection datasets as shown in Appendix.

## 5 Conclusion

This paper presents KGD, a novel knowledge distillation technique that exploits the implicit KG of CLIP to adapt large-vocabulary object detectors for handling various unlabelled downstream data. KGD consists of two consecutive stages including KG extraction and KG encapsulation which extract and encapsulate visual and textual KGs simultaneously, thereby providing complementary vision and language knowledge to facilitate unsupervised adaptation of large-vocabulary object detectors towards various downstream detection tasks. Extensive experiments on multiple widely-adopted detection datasets demonstrate that KGD consistently outperforms state-of-the-art techniques by clear margins.

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

# A   Appendix

## A.1   Datasets

We perform experiments on 11 object detection datasets that span different downstream domains including the object detection for autonomous driving [73, 74], autonomous driving under different weather and time-of-day conditions [75], intelligent surveillance [80, 81, 82], common objects [78, 79], and artistic illustration [83]. We provide dataset details here.

**Cityscapes [73]** is a dataset designed for the purpose of understanding street scenes. It comprises images captured in 50 different cities, encompassing a total of 2975 training images and 500 validation images. These images are captured under normal weather conditions with pixel-wise instance annotations of 8 categories.

**Vistas [74]** is an autonomous driving dataset collected for street scene understanding. It comprises a vast collection of high-resolution images that encompass diverse urban environments from various locations worldwide. The dataset consists of 18000 training images and 2000 validation images with pixel-wise instance annotations.

**BDD100k [75]** is a large-scale driving video dataset with a wide range of diverse driving scenarios. It comprises various weather conditions such as clear, cloudy, overcast, rainy, snowy, and foggy, as well as different times of the day including dawn, daytime, and night. The dataset contains 70000 training videos and 10000 validation videos. Each video is annotated with bounding boxes for objects of 10 distinct categories.

**MIO-TCD [80]** is an intelligent surveillance dataset collected for traffic analysis. It comprises 137743 images captured at different times of the day and various periods throughout the year. The images are captured from diverse viewing angles. Each image in the dataset is annotated with bounding boxes, providing precise spatial locations of objects of 11 categories.

**BAAI [81]** is a dataset collected for surveillance applications. It comprises 5000 high-quality images captured by the VANJEE smart base station positioned at a height of 4.5 meters. Each image in the dataset is annotated with bounding boxes, providing spatial locations of objects of 12 categories.

**VisDrone [82]** is a surveillance dataset captured using drone-mounted cameras in different scenarios, and under various weather and lighting conditions. It comprises 288 video clips with 261908 frames, as well as an additional set of 10209 static images. These frames and images are annotated with more than 2.6 million bounding boxes of objects of 10 categories.

**Pascal VOC [78]** consists of two distinct sub-datasets: Pascal VOC 2007 and Pascal VOC 2012. The former comprises a total of 2501 training images and 2510 validation images, while the latter encompasses a larger set of 5717 training images and 5823 validation images. This dataset includes bounding box annotations of 20 object categories.

**Objects365 [79]** is a large-scale object detection dataset with 2 million images, 30 million bounding boxes, and 365 categories, which is designed for detecting diverse objects in the wild.

**Clipart1k [83]** is a prominent dataset employed in cross-domain object detection, comprising 1000 clipart images collected from one dataset (CMPlaces [87]) and two image search engines (Openclipart and Pixabay). Each image in the dataset is annotated with bounding boxes for objects that share 20 categories with Pascal VOC [78].

**Watercolor2k [83]** comprises a collection of 2000 watercolor images with image and instance-level annotations of 6 categories. It is also a prominent dataset employed in cross-domain object detection.

**Comic2k [83]** contains 2000 comic images with image and instance-level annotations, sharing 6 categories with Pascal VOC [78].

## A.2   Implementation details

We adopt Detic [3] as LVD, where CenterNet2 [22] with Swin-B [88] is pre-trained on LVIS [9] for object localization and ImageNet-21K [89] for object classification. During adaption, the updating

---

https://openclipart.org/
https://pixabay.com/

**Algorithm 1:** Domain Adaptation for Large-Vocabulary Object Detectors

---

**Input:** unlabeled downstream data $\mathcal{D}_t$, pretrained LVD $\Phi$, CLIP image encoder $V$, CLIP text encoder $T$, WordNet database retrieval function WNRetrieve;

**Output:** domain adaptive detector $\Phi_s$;

1 Initialization: teacher detector $\Phi_t \leftarrow \Phi$, student detector $\Phi_s \leftarrow \Phi$, maximum iteration $l$, momentum updating frequency $t_{mom}$, momentum updating rate $\mu$;

2 Extract LKG by Eq.(2)-(4);

3 Extract VKG by Eq.(10);

4 **for** $t \leftarrow 0$ **to** $l$ **do**

5      Sample a batch of $B$ targe domain samples: $\{\mathbf{x}_b^t\}_{b=1}^B \subseteq \mathcal{D}_t$;

6      Generate pseudo label set $\{\hat{\mathbf{y}}_b\}_{b=1}^B$ by Eq.(5);

7      Generate CLIP feature matrix set $\{F_b\}_{b=1}^B$ with Eq.(6);

8      Encapsulate LKG by Eq.(9);

9      Encapsulate VKG by Eq.(14);

10      Minimize overall objective function Eq.(15) by updating $\Phi_s$ and GCN; Update VKG by Eq.(12) and (13);

11      **if** $t \% t_{mom} == 0$ **then**

12          Update EMA detector: $\Phi_t \leftarrow \mu\Phi_t + (1-\mu)\Phi_s$;

---

rate of EMA detector is set as 0.9999. The pseudo labels generated by the teacher detector with confidence greater than the threshold $\tau = 0.25$ are selected for adaptation. We use AdamW [90] optimizer with initial learning rate $5 \times 10^{-6}$ and weight decay $10^{-4}$, and adopt a cosine learning rate schedule without warm-up iterations. The batch size is 2 and the image's shorter side is set to 640 while maintaining the aspect ratio unchanged.

### A.3 Algorithm of KGD

We describe the detailed algorithm of our proposed KGD in Algorithm 1.

### A.4 Additional Discussion

#### A.4.1 Parameter Study

**The value of $\lambda$:** In the Eq. (12), the nodes of VKG are preliminarily updated with a pre-defined $\lambda$. $\lambda$ is set as 0.9999. We studied $\lambda$ by changing it from 0.99 to 0.999999. The table below reports the experiments over the Cityscapes dataset. It shows that both an excessivel small $\lambda$ or excessively large $\lambda$ lead to performance degradation, largely because an excessively small $\lambda$ (i.e., 0.99) introduces more noise and fluctuation, while an excessively large $\lambda$ (i.e., 0.999999) results in a sluggish response to the latest data changes, failing to update VKG nodes promptly. However, an appropriate value (0.9999) of $\lambda$ can suppress noise and data fluctuation while promptly updating VKG nodes to timely respond to the latest data distribution shift.

Table 9: Parameter analysis of KGD for $\lambda$.

| $\lambda$ | 0.99 | 0.999 | 0.9999 | 0.99999 | 0.999999 |
|---|---|---|---|---|---|
| AP50 | 49.9 | 51.5 | 53.6 | 52.3 | 51.8 |

**The value of $\alpha$:** Eq. (13) incorporate the downstream visual graph knowledge into VKG with a pre-defined $\alpha$. $\alpha$ is the scaling factor used to control the weights of the neighboring node features and the node's own features during this process. $\alpha$ is set as 0.001. We studied $\lambda$ by changing it from 0.001 to 1.0. The table below reports the experiments over the Cityscapes dataset. It shows that both an too small $\lambda$ or too large $\alpha$ lead to performance degradation, largely because a too small $\alpha$ may cause the model to fail to effectively utilize the information from neighboring nodes, thus not fully capturing the structure of the graph and the relationships between nodes, while a too large $\alpha$ may cause noise to propagate through the graph, making the node updates more susceptible to outliers or noisy data.

Table 10: Parameter analysis of KGD for $\alpha$.

| $\alpha$ | 0.0001 | 0.001 | 0.01 | 0.1 | 1 |
|---|---|---|---|---|---|
| AP50 | 50.9 | 51.0 | 53.6 | 49.9 | 49.2 |

Table 11: Ablation studies of KGD with Language Knowledge Graph Distillation (KGD-L) and Vision Knowledge Graph Distillation (KGD-V). The experiments are conducted on the Cityscapes, BAAI, VOC, and Clipart1k.

| Method | Language Knowledge Graph Distillation | Vision Knowledge Graph Distillation | AP50 | | |
|---|---|---|---|---|---|
| | | | BAAI | VOC | Clipart1k |
| Detic [3] (Baseline) | | | 20.6 | 83.9 | 61.0 |
| | ✓ | | 22.2 | 86.1 | 66.5 |
| | | ✓ | 22.4 | 86.2 | 67.1 |
| **KGD (Ours)** | ✓ | ✓ | **24.3** | **86.9** | **69.1** |

### A.4.2 Additional ablation studies

We have conducted additional ablation study on 3 object detection datasets that span different downstream domains including the object detection for intelligent surveillance (BAAI), common objects (VOC), and artistic illustration (Clipart1k). As Table 11 shows, the behavior consistent across datasets that span different downstream domains.

### A.4.3 Combination of language knowledge graph extraction and vision knowledge graph strategies

We conducted experiments and reports the results of using "LKG Extraction with category name" and "Static VKG Extraction" jointly in Table 12. As a comparison, our proposed KGD shows clear improvements as the language and vision information extracted along the training process dynamically stabilizes and improves the model adaptation, validating the performance gain largely comes from our novel KGD designs instead of merely using "LKG Extraction with category name" and "Static VKG Extraction".

### A.4.4 Comparisons with semi-supervised learning.

We would clarify that the motivation for using knowledge graphs is to explicitly and comprehensively extract CLIP knowledge for effectively de-noising pseudo labels generated by LVDs when adapting LVDs. On the other hand, directly utilizing CLIP to obtain pseudo-labels could also benefit unsupervised domain adaptation of LVDs, but it may be less effective. The reason lies in that knowledge graphs carry not only the information of each category but also inter-class relations, while pseudo-labels only carry the former information. As the table bleow, we conduct new experiments that adapt Detic with semi-supervised learning [35] using CLIP-generated pseudo-labels. The experimental results show that our proposed KGD outperforms the semi-supervised learning using CLIP-generated pseudo-labels, validating the performance gain largely comes from our novel KGD designs instead of merely using pseudo-labels from CLIP.

### A.4.5 Comparisons with other pseudo label generation methodologies.

Our proposed KGD generates the pseudo labels (PLs) in real-time during training as discussed in the paper. We examine the proposed KGD by comparing it with four strategies, i.e., "Offline Detic Generated PLs", "Offline CLIP Generated PLs", "Online VL-PLM [91] Generated PLs", and "Online RegionCLIP [92] Generated PLs".

For "Offline Detic Generated PLs", the pseudo labels (PLs) are generated offline for all downstream samples using pretrained Detic [3], and remain unchanged during adaptation. For "Offline CLIP [7] Generated PLs", pretrained Detic [3] generates instance bounding box pseudo labels (PLs) offline while CLIP [7] generates instance category pseudo label offline, and remains unchanged during adaptation. "Online VL-PLM [91] Generated PLs" and "Online RegionCLIP [92] Generated PLs" refer to generate pseudo labels (PLs) from Detic [3] using other pseudo label generation methods including VL-PLM [91] and RegionCLIP [92], respectively. As Table 14 shows, all the compared

Table 12: Combination of language knowledge graph extraction and vision knowledge graph strategies. The experiments are conducted on the Cityscapes.

| Method | Detic [3] (Source only) | | | | KGD |
|---|---|---|---|---|---|
| LKG Extraction with category names | | ✓ | | ✓ | |
| Static VKG Extraction | | | ✓ | ✓ | |
| LKG Extraction with WordNet Hierarchy | | | | | ✓ |
| Dynamic VKG Extraction | | | | | ✓ |
| AP50 | 46.5 | 51.9 | 51.9 | 52.4 | **53.6** |

Table 13: Study of different adaptation strategies for LVDs on Cityscapes dataset [73].

| Method | AP50 |
|---|---|
| Detic [3] (Baseline) | 46.5 |
| semi-supervised learning [35] (using CLIP-generated pseudo-labels) | 48.8 |
| **KGD (Ours)** | **53.6** |

strategies achieve sub-optimal performance. The offline generated pseudo labels in both "Offline Detic Generated PLs" and "Offline CLIP Generated PLs" are noised and degrade the unsupervised domain adaptation performance. "Online VL-PLM [91] Generated PLs" and "Online RegionCLIP [92] Generated PLs" denoise the online generated pseudo labels when adapting LVD, which benefit the unsupervised domain adaptation, but it may be less effective. The reason lies in that knowledge graphs carry not only the information of each category but also inter class relations, while pseudo labels (PLs) only carry the former information. The experimental results show that our proposed KGD outperforms the compared methods, validating the performance gain largely comes from our novel KGD designs instead of merely using pseudo-labels from Detic [3] and CLIP [7].

Table 14: Study of different adaptation strategies for LVDs on Cityscapes dataset [73].

| Method | AP50 |
|---|---|
| Detic [3] (Baseline) | 46.5 |
| Offline Detic Generated PLs | 48.5 |
| Offline CLIP Generated PLs | 50.1 |
| Online VL-PLM Generated PLs | 51.5 |
| Online RegionCLIP Generated PLs | 51.9 |
| **KGD (Ours)** | **53.6** |

#### A.4.6 Comparisons with prior knowledge graph-related distillation methods.

We conduct experiments to compare our KGD with prior knowledge graph-related methods [93, 94]. The results in Table 15 and 16 show that our KGD outperform KGE [93] and Context Matters [94] clearly, largely becuase the knowledge graphs in KGE [93] and Context Matters [94] are hand-crafted by domain experts while ours is built and learnt from CLIP.

#### A.4.7 Distance metrics for constructing knowledge graph.

We explore the feature distance metrics for constructing knowledge graphs. We conduct experiments that construct knowledge graphs with the following feature distance metrics: 1) Cosine Similarity [95], 2) Euclidean Distance [95], 3) Manhattan Distances [95]. The results in Table 17 show that our KGD works effectively and consistently with different feature distance metrics. Besides, the cosine similarity metric performs the best, largely because CLIP is also trained with cosine similarity where using the same metric to distill its knowledge works the best reasonably.

Table 15: Benchmarking over autonomous driving datasets under various weather and time conditions. We adopt AP50 in evaluations. The results of all methods are acquired with the same baseline [3] as shown in the first row.

| Method | Cityscapes [73] | Vistas [74] | BDD100K-weather [75] | | | | | BDD100K-time-of-day [75] | | |
|---|---|---|---|---|---|---|---|---|---|---|
| | | | rainy | snowy | overcast | cloudy | foggy | daytime | dawn&dusk | night |
| Detic [3] (Baseline) | 46.5 | 35.0 | 34.3 | 33.5 | 39.1 | 42.0 | 28.4 | 39.2 | 35.3 | 28.5 |
| KGE [93] | 48.9 | 36.0 | 35.5 | 34.4 | 40.5 | 41.2 | 29.7 | 40.1 | 36.6 | 29.0 |
| Context Matters [94] | 49.4 | 36.6 | 36.3 | 35.0 | 41.7 | 42.4 | 30.2 | 41.5 | 37.2 | 29.7 |
| **KGD (Ours)** | **53.6** | **40.3** | **37.3** | **37.1** | **44.6** | **48.2** | **38.0** | **46.6** | **41.0** | **31.2** |

Table 16: Benchmarking over common objects datasets, intelligent surveillance datasets, and artistic illustration datasets. We adopt AP50 in evaluations. The results of all methods are acquired with the same baseline [3] as shown in first row.

| Method | Common Objects | | Intelligent Surveillance | | | Artistic Illustration | | |
|---|---|---|---|---|---|---|---|---|
| | VOC [78] | Objects365 [79] | MIO-TCD[80] | BAAI [81] | VisDrone [82] | Clipart1k [83] | Watercolor2k [83] | Comic2k [83] |
| Detic [3] (Baseline) | 83.9 | 29.4 | 20.6 | 20.6 | 19.0 | 61.0 | 58.9 | 51.2 |
| KGE [93] | 85.4 | 31.2 | 20.3 | 23.5 | 19.4 | 62.4 | 58.1 | 50.5 |
| Context Matters [94] | 85.9 | 31.7 | 20.9 | 23.3 | 19.9 | 62.9 | 59.1 | 52.3 |
| **KGD (Ours)** | **86.9** | **34.4** | **24.6** | **24.3** | **23.7** | **69.1** | **63.5** | **55.6** |

### A.4.8 Training and inference overhead analysis.

We study the training and inference time of all the compared methods, and Table 18 shows the results on Cityscapes. † signifies that the methods employ WordNet to retrieve category descriptions given category names, and CLIP to predict classification pseudo labels for objects. The experiments are conducted on one RTX 2080Ti. It shows that incorporating CLIP into unsupervised domain adaptation introduces a few additional overhead on training time and almostly does not affect inference time. The reason lies in that the cropped object regions are processed by CLIP in a parallel manner during training while the inference pipeline does not involve CLIP. Besides, we compare the memory usage and computational overhead with other methods in the table below, where It can be seen that while the involvement of CLIP during training increases memory usage and computational overhead due to the processing of cropped object regions, the memory usage and computational overhead during inference remain comparable to baseline methods. This is because the inference pipeline does not involve CLIP, thus maintaining efficiency and ensuring that its practicability for deployment.

The Proposal Network of Faster R-CNN generates a large number of region proposals on the input image (i.e., thousands to tens of thousands of region proposals), which make VILD [84]-like methods very slow. On the other hand, our KGD is performed only on the selected box predictions (i.e., the box predictions after the confidence thresholding), where the number of involved predictions is much smaller (i.e., a few to several dozen), which only introduces a few additional computation overhead. In another word, Eq. (6) in our manuscript works by cropping the selected box predictions (i.e., the pseudo labels after the prediction filtering and thresholding), instead of cropping all region proposals as in VILD [84], which significantly reduces the number of regions to be cropped and is much more efficient. We validate above statements by examining the training and inference time of all the compared methods, as shown in Table 18. It shows that the operation of cropping object regions and using CLIP introduces a few additional computation overhead in training time and almost does not affect the inference time. The reason lies in that we only crop a limited number of object regions (i.e., selected ones) and process them with CLIP model in a parallel manner during training, while the inference pipeline does not involve these procedures.

### A.4.9 Study of vocabulary size of KGD.

The construction of knowledge graphs greatly improves the quality of pseudo-labels. We study how vocabulary size affects the cost of model training and inference. As shown in Table 19, the increase in vocabulary size of knowledge graphs brings little computation overhead, largely because our knowledge graphs are implemented in a efficient computation manner.

### A.4.10 Experiments with open-vocabulary detectors.

The open-vocabulary detector (OVD) aims to detect objects in novel categories described by text inputs [84]. Similar to Large-vocabulary Detectors (LVDs), OVDs also suffer from domain dis-

Table 17: Study of different distance metrics for constructing KG. The experiments are conducted on the Cityscapes dataset.

| Distance Metrics | Cosine Similarity | Euclidean Distance | Manhattan Distances |
|---|---|---|---|
| AP50 | 53.6 | 52.9 | 53.1 |

Table 18: Training and inference time analysis of all the compared methods. The experiments are conducted on one RTX 2080Ti. † signifies that the methods employ WordNet to retrieve category descriptions given category names, and CLIP to predict classification pseudo labels for objects.

| Method | MT [45] | MT [45]† | SHOT [44] | SHOT [44]† | SFOD [46] | SFOD [46]† | HCL [50] | HCL [50]† | IRG-SFDA [51] | IRG-SFDA [51]† | KGD (Ours) |
|---|---|---|---|---|---|---|---|---|---|---|---|
| Training Time (hours) | 4.083 | 5.022 | 4.055 | 5.045 | 4.110 | 5.193 | 4.133 | 5.095 | 4.158 | 5.222 | 5.267 |
| Training Memory Usage (MB) | 3219 | 3219 | 7245 | 3219 | 7245 | 3219 | 7245 | 3219 | 7245 | 3219 | 7245 |
| Training Computational Overhead (GFLOPs) | 21.74 | 21.74 | 42.39 | 21.74 | 42.39 | 21.74 | 42.39 | 21.74 | 42.39 | 21.74 | 42.39 |
| Inference Speed (images per second) | 6.700 | 6.767 | 6.749 | 6.809 | 6.523 | 6.752 | 6.689 | 6.683 | 6.758 | 6.701 | 6.758 |
| Inference Memory Usage (MB) | 3219 | 3219 | 3219 | 3219 | 3219 | 3219 | 3219 | 3219 | 3219 | 3219 | 3219 |
| Inference Computational Overhead (GFLOPs) | 21.74 | 21.74 | 21.74 | 21.74 | 21.74 | 21.74 | 21.74 | 21.74 | 21.74 | 21.74 | 21.74 |

crepancies when applying to the downstream dataset, because their training data often exhibits different distributions and vocabularies as compared with the downstream data. We investigate how our proposed KGD works with open-vocabulary detectors by conducting experiments, as shown in Table 20. It can be observed that our proposed KGD can also improve the performance of OVDs significantly, validating the generalization ability of our KGD on different detectors.

### A.4.11 Generalization across different datasets.

We study the generalization of our KGD by conducting domain-adaptive object detection on 11 widely studied object detection datasets. Table 21 summarizes the detection results averaged on 11 datasets, i.e., Cityscapes [73], Vistas [74], BDD100k [75], MIO-TCD [80], BAAI [81], VisDrone [82], Pascal VOC [78], Objects365 [79], Clipart1k [83], Watercolor2k [83], and Comic2k [83]. It shows that our KGD outperforms the state-of-the-art methods clearly on 11 datasets.

### A.4.12 Motivation analysis.

Our motivation arises from the observation that the LVD pretrained on the source domain can accurately locate objects in various downstream-domain images but struggles with classifying these located objects [3]. To analyze this motivation, we disentangle object detection task into two sub-tasks, i.e., object locating and object classification, and evalute them respectively by introducing two new types of metrics: Category-agnostic AP50 and Ground Truth (GT) bounding box-corrected AP50. In Category-agnostic AP50, we correct the object classification predictions (i.e., replace the predicted object categories with ground truth object categories) before the conventional AP50 evaluation, aiming to assess the accuracy of object locating sub-task only. In GT bounding box-corrected AP50, we correct the object locating predictions (i.e., replace the predicted object bounding boxes with ground truth bounding boxes) before the conventional AP50 evaluation, aiming to assess the accuracy of object classification sub-task only.

Table 22 reports the results of Detic [3] over the Cityscapes dataset, which are measured in AP50, Category-agnostic AP50, and GT bounding box-corrected AP50, respectively. We can observe that introducing GT category and bounding box information to correct the predictions bring 14.3% and 4.6% improvements respectively. It shows that the performance degradation of Detic [3] on downstream domains largely comes from misclassification prompts as compared with the mislocating issues, which is well-aligned with our motivation.

### A.4.13 Study Limitations

The proposed KGD uses Detic as the LVD which is pre-trained on LVIS [9] for object localization and ImageNet-21K [89] for object classification. When adapting LVD to datasets of different domains especially from different scenarios, there is a risk that the detector will fail to localize certain categories of objects that are not included in ImageNet-21K and LVIS. In our future work, we intend to address this issue by employing larger pre-training dataset or transferring the more comprehensive knowledge of Multi-Modal Large Language Models.

Table 19: Study of vocabulary size of knowledge graphs. The experiments are conducted on one RTX 2080Ti.

| Dataset | Vocabulary Size | Training Time (hours) | Inference Speed (images per second) |
|---|---|---|---|
| Watercolor2k | 6 | 5.165 | 6.696 |
| Comic2k | 6 | 5.159 | 6.721 |
| Cityscapes | 8 | 5.167 | 6.718 |
| Vistas | 8 | 5.163 | 6.694 |
| VisDrone | 10 | 5.169 | 6.720 |
| BDD100k | 10 | 5.168 | 6.788 |
| MIO-TCD | 11 | 5.167 | 6.677 |
| BAAI-VANJEE | 12 | 5.169 | 6.714 |
| Clipart1k | 20 | 5.165 | 6.721 |
| Pascal VOC | 20 | 5.168 | 6.698 |
| Objects365 | 365 | 5.171 | 6.723 |

Table 20: Experiments with Open-Vocabulary Detectors over Cityscapes dataset. We adopt AP50 in evaluations. We can observe that our proposed KGD can also improve the performance of OVDs (e.g., GLIP [96], VILD [84], RegionKD [85], UniDet [97], and RegionCLIP [92]) significantly, validating the generalization ability of our KGD on different detectors.

| Method | AP50 |
|---|---|
| Detic [3] [3] (Baseline) | 46.9 |
| Detic [3]+KGD | 53.6 |
| GLIP [96] (Baseline) | 44.7 |
| GLIP [96]+KGD | 52.1 |
| VILD [84] (Baseline) | 45.5 |
| VILD [84]+KGD | 53.1 |
| RegionKD [85] (Baseline) | 48.5 |
| RegionKD [85]+KGD | 54.3 |
| UniDet [97] (Baseline) | 52.6 |
| UniDet [97]+KGD | 54.7 |
| RegionCLIP [92] (Baseline) | 50.1 |
| RegionCLIP [92]+MT | 51.9 |
| RegionCLIP [92]+SHOT | 51.6 |
| RegionCLIP [92]+SFOD | 50.9 |
| RegionCLIP [92]+HCL | 51.2 |
| RegionCLIP [92]+IRG-SFDA | 52.2 |
| RegionCLIP [92]+KGD | 55.4 |

## A.5   More Qualitative Comparisons

We provide qualitative illustrations of KGD over downstream datasets. As shown in Figure 3-7, KGD produces accurate detection across multiple datasets, demonstrating its capability to adapt LVDs to various downstream domains of very different data distribution and vocabulary.

Table 21: Average results over 11 widely studied datasets. † signifies that the methods employ WordNet to retrieve category definitions given category names, and CLIP to predict classification pseudo labels for objects. The results of all methods are acquired with the same baseline [3] as shown in the first column.

| Method | Detic [3] (Baseline) | MT [45] | MT [45]† | SHOT [44] | SHOT [44]† | SFOD [46] | SFOD [46]† | HCL [50] | HCL [50]† | IRG-SFDA [51] | IRG-SFDA [51]† | KGD (Ours) |
|---|---|---|---|---|---|---|---|---|---|---|---|---|
| AP50 | 39.24 | 39.91 | 40.95 | 40.21 | 41.47 | 40.14 | 41.50 | 40.43 | 41.67 | 40.93 | 42.12 | 44.44 |

Table 22: Benchmarking Detic over Cityscapes [73] dataset with AP50, Category-agnostic AP50, and GT bounding box-corrected AP50.

| Metric | AP50 | Category-agnostic AP50 | GT bounding box-corrected AP50 |
|---|---|---|---|
| Detic [3] | 46.9 | 61.2 (+14.3) | 51.5 (+4.6) |

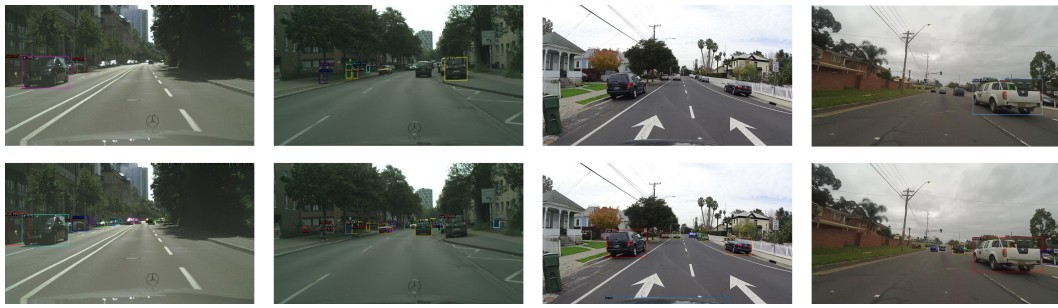

Figure 3: Qualitative comparisons over autonomous-driving data. Zoom in for details. Top: Detic [3]. Bottom: KGD (Ours).

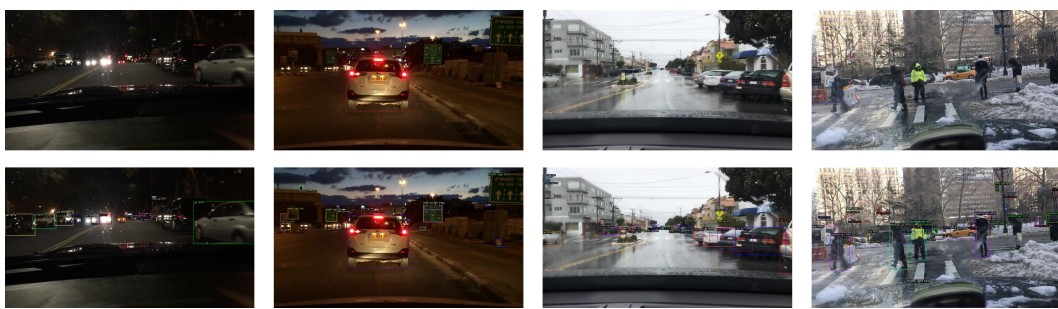

Figure 4: Qualitative comparisons over autonomous-driving data under different weather and time-of-day conditions. Zoom in for details. Top: Detic [3]. Bottom: KGD (Ours).

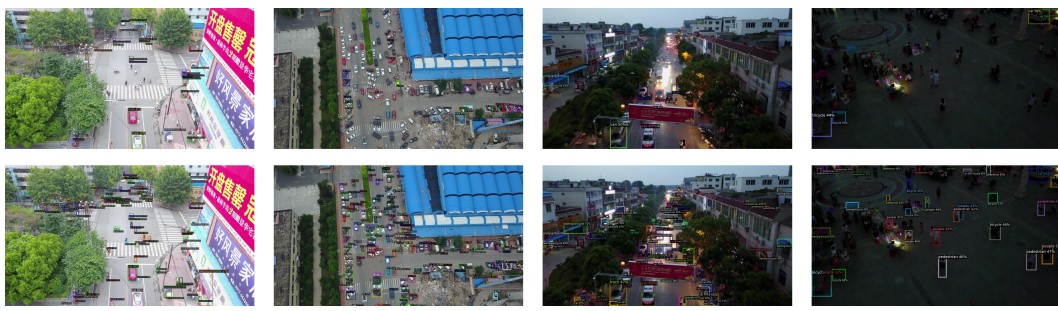

Figure 5: Qualitative comparisons over intelligent-surveillance data. Zoom in for details. Top: Detic [3]. Bottom: KGD (Ours).

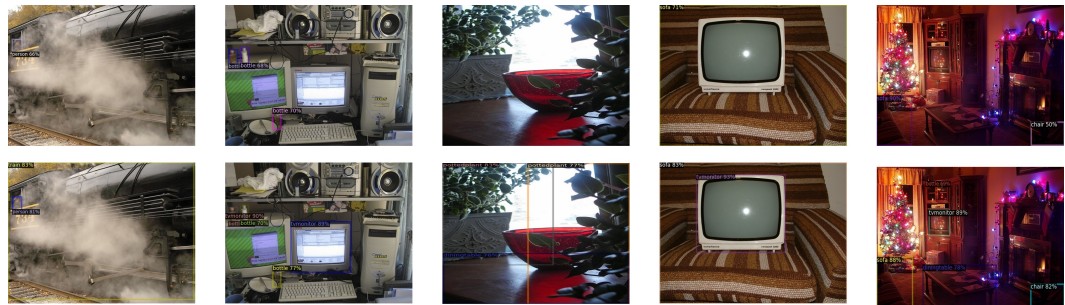

Figure 6: Qualitative comparisons over common-object data. Zoom in for details. Top: Detic [3]. Bottom: KGD (Ours).

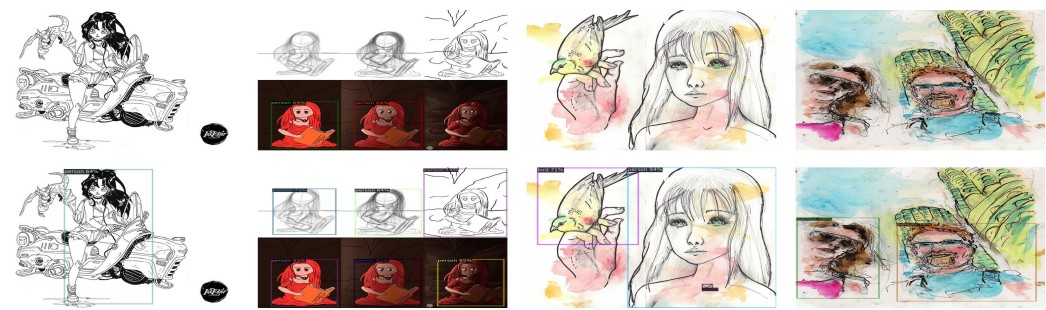

Figure 7: Qualitative comparisons over artistic illustration data. Zoom in for details. Top: Detic [3]. Bottom: KGD (Ours).

