# OpenReview forum: "Domain Adaptation for Large-Vocabulary Object Detectors"
_NeurIPS.cc/2024/Conference — NeurIPS 2024 poster_

### Official Review · Reviewer_5mbu · 2024-07-08

**Soundness:** 3
**Presentation:** 2
**Contribution:** 3
**Rating:** 5
**Confidence:** 4

**Summary:**

This paper addresses the domain generalization problem of large-vocabulary object detectors. Without requiring additional annotations, it uses the Knowledge Graph Distillation technique to transfer knowledge from CLIP, enhancing the detector's generalization capability on downstream datasets.

**Strengths:**

1. Strong performance across 11 widely used dataset.
2. Detailed implementation details, which I really appreciate, are provided in the appendix. It includes thorough analysis and explanation from the motivation to the final results.
3. The proposed method is highly intuitive and easy to understand.

**Weaknesses:**

1. Speed: Although the appendix provides training times, considering that Equation 6 requires cropping proposals from the detector and sending them into CLIP, this step is very time-consuming (as seen in VILD[1], which took almost a few days on the COCO dataset). Given that the training set includes much larger datasets like Object365, how was this training time calculated?
2. Performance: As shown in VILD and other open-vocabulary papers, adding CLIP scores improves detection performance on novel classes. Considering this paper also uses CLIP to assist in out-of-domain detection, and the compared methods do not use CLIP, directly embedding CLIP scores as a baseline to highlight the necessity of GCN is essential.

[1] Open-vocabulary Object Detection via Vision and Language Knowledge Distillation

**Questions:**

see weakness.

**Limitations:**

yes, the authors has adequately discussed the potential negative societal impact of their work and limitations.

---

> ### Author Rebuttal · Authors · 2024-08-07
>
> **Response to Weaknesses-1**:
>
> The Proposal Network of Faster R-CNN generates a large number of region proposals on the input image (i.e., thousands to tens of thousands of region proposals), which make VILD-like methods very slow. On the other hand, our KGD is performed only on the selected box predictions (i.e., the box predictions after the confidence thresholding), where the number of involved predictions is much smaller (i.e., a few to several dozen), which only introduces a few additional computation overhead. In another word, Eq. (6) in our manuscript works by cropping the selected box predictions (i.e., the pseudo labels after the prediction filtering and thresholding), instead of cropping all region proposals as in VILD [1], which significantly reduces the number of regions to be cropped and is much more efficient.
>
> We validate above statements by examining the training and inference time of all the compared methods, as shown in Table 11 in our manuscript. It shows that the operation of cropping object regions and using CLIP introduces a few additional computation overhead in training time and almost does not affect the inference time. The reason lies in that we only crop a limited number of object regions (i.e., selected ones) and process them with CLIP model in a parallel manner during training, while the inference pipeline does not involve these procedures.
>
>
> **Response to Weaknesses-2**:
>
> Thank you for your suggestions!
> We would clarify that we have introduced the WordNet and CLIP into the compared methods for fair comparison, which are signified by $\star$ as shown in Table 1 and 2 in the manuscript. Specifically, the methods employ WordNet to retrieve category descriptions given category names, and CLIP to predict classification pseudo labels (CLIP scores) for objects. The experiments results show that KGD still outperforms the state-of-the-art clearly when CLIP and WordNet are incorporated into the compared methods, validating that the performance gain largely comes from our designed KGD method (including the GCN) instead of merely using CLIP scores. Moreover, we would like to clarify that we compared our KGD with existing CLIP knowledge distillation methods designed for detection tasks (including VILD, RegionKD, and OADP) in Table 4 in the manuscript.
>
> Table 4 in the manuscript reports the experimental results over the Cityscapes dataset, which shows existing CLIP knowledge distillation methods do not perform well in adapting LVDs to downstream tasks, while our KGD works for LVDs adaption effectively, which further validate that the performance gain largely comes from our designed KGD method (including the GCN) instead of merely using CLIP scores.

---

> > ### Comment · Reviewer_5mbu · 2024-08-13
> >
> > I appreciate the author's efforts in addressing all of concerns during rebuttal - most of which with satisfactory explanations. I currently intend to keep the rating positive.

---

### Official Review · Reviewer_1Ck4 · 2024-07-12

**Soundness:** 3
**Presentation:** 3
**Contribution:** 3
**Rating:** 5
**Confidence:** 5

**Summary:**

This paper highlights the problem that in domain adaptation, detectors often correctly localize but misclassify. To solve this problem, this paper proposes the Knowledge Graph Distillation method, which uses the pre-trained knowledge of VLM to supplement the relational knowledge of various categories in vision and language required for object detection. Experiments over multiple benchmarks validate the effectiveness.

**Strengths:**

1 The proposed method is easy to follow.

2 The proposed method exploits the prior knowledge of VLM to annotate unlabeled domains through knowledge graph distillation, which is a promising and efficient direction.

3 Experiments are conducted to verify the effectiveness of the proposed method.

**Weaknesses:**

1 Computation overhead: Although the paper shows the training time and inference speed compared with other methods in Table 11, it does not compare memory usage and computational overhead.

2 Insufficient and unfair experimental comparison. The proposed KGD utilizes the strong generalization capability of the visual-language model (VLM) in the region (proposal). Therefore, it is unfair to compare directly with the traditional source-free domain adaptation method as they did not make any special design for VLM. The proposed KGD should be compared with methods based on visual-language models, such as RegionCLIP, which also uses the idea of ​​regional distillation. However, the comparison with methods such as RegionCLIP is only conducted on the Cityscapes dataset (Table 9), and the improvement is not particularly obvious (1.7%).

**Questions:**

1 I would like to know, if traditional methods (such as MT, etc.) are applied to methods like RegionCLIP, what is the performance gap with the proposed KGD?

**Limitations:**

Yes

---

> ### Author Rebuttal · Authors · 2024-08-07
>
> **Response to Weakness-1:**
> Thanks for your comments.
> As suggested, we compare the memory usage and computational overhead with other methods in the table below, where $\star$ signifies that the methods employ WordNet to retrieve category descriptions given category names, and CLIP to predict classification pseudo labels for objects. The experiments are conducted on one RTX 2080Ti.
> It can be seen that while the involvement of CLIP during training increases memory usage and computational overhead due to the processing of cropped object regions, the memory usage and computational overhead during inference remain comparable to baseline methods. This is because the inference pipeline does not involve CLIP, thus maintaining efficiency and ensuring that its practicability for deployment.
>
> Method | MT | MT$\star$ | SHOT | SHOT$\star$| SFOD |SFOD$\star$ |HCL  |HCL$\star$  |IRG-SFDA |IRG-SFDA $\star$| KGD (Ours)
> -|-|-|-|-|-|-|-|-|-|-|-
> training |Memory Usage (MB) |3219 |7245 |3219 |7245 |3219 |7245 |3219 |7245 |3219 |7245 |7245
> training |Computational overhead (GFLOPs) |21.74 |42.39 |21.74 |42.39 |21.74 |42.39 | 21.74 | 42.39 | 21.74 | 42.39 | 42.39
> inference |Memory Usage (MB)| 3219| 3219 |3219 |3219 |3219 |3219 | 3219| 3219 |3219 |3219 |3219
> inference |Computational overhead (GFLOPs) |21.74 |21.74| 21.74 |21.74 |21.74 |21.74 |21.74 |21.74 |21.74 |21.74 |21.74
>
> **Response to Weakness-2:**
>
> Many thanks for your suggestions!
> We would like to clarify that we introduced the WordNet and CLIP into the compared methods for fair comparison, which are signified by $\star$ as shown in Table 1 and 2 in the manuscript.
> Specifically, for the methods signified by \dag, we employ WordNet to retrieve category descriptions given category names, and CLIP to predict classification pseudo labels for objects.
> The experiments results show that KGD still outperforms the state-of-the-art clearly when CLIP and WordNet are incorporated into the compared methods, validating that the performance gain largely comes from our novel KGD (including the GCN) instead of merely using CLIP and WordNet.
>
> **Response to Question-1:**
> As suggested, we conducted the experiment of applying traditional methods (Mean Teacher(MT) [53], SHOT [29], SFOD [27], HCL [18], and IRG-SFDA [55]) on RegionCLIP.
> As the table below shows, KGD performs consistently well on this scenario.
> We will include the new experiments into the updated paper later. Thank you for your suggestion!
>
> Method |AP50
> -|-
> RegionCLIP |50.1
> RegionCLIP+MT |51.9
> RegionCLIP+SHOT|51.6
> RegionCLIP+SFOD|50.9
> RegionCLIP+HCL|51.2
> RegionCLIP+IRG-SFDA|52.2
> KGD|53.6

---

> > ### Comment · Reviewer_1Ck4 · 2024-08-13
> >
> > Thanks for the authors' response. I will raise my score to 5.

---

### Official Review · Reviewer_Jwgx · 2024-07-12

**Soundness:** 2
**Presentation:** 2
**Contribution:** 2
**Rating:** 5
**Confidence:** 4

**Summary:**

This paper addresses the challenges faced by Large-vocabulary object detectors (LVDs) in recognizing objects across diverse categories, particularly due to domain discrepancies in data distribution and object vocabulary. The paper proposes a novel approach named Knowledge Graph Distillation (KGD) that leverages the implicit knowledge graphs (KG) within the CLIP model to enhance the adaptability of LVDs to various downstream domains. The paper validated the effectiveness on a wide range of datasets.

**Strengths:**

1. The task tackled in this paper is highly significant because successfully generalizing Large-vocabulary detectors (LVDs) to downstream tasks can substantially increase the practical utility of these models.
2. The paper presents a detailed description of the method and offers numerous formal justifications and clarifications.

**Weaknesses:**

1. The paper has not clearly explained the motivation for using knowledge graphs as a tool. If the goal is to employ LVDs for various downstream tasks involving unlabeled data, one potential approach could be to use CLIP to obtain pseudo-labels and then incrementally update the LVDs through semi-supervised learning. In this scenario, it appears that a specifically designed knowledge graph may not be necessary.

2. The methods used for comparison in the experiments are outdated, with no works from 2023 or 2024 included. This makes it difficult to adequately demonstrate the effectiveness of the proposed method.

3. The knowledge graph distillation technique is likely based on prior research in the field. From a design perspective, it would be beneficial if the paper could emphasize the improvements the proposed method offers over previous approaches in this area.

4. The ablation experiments are not sufficiently comprehensive. It would be advantageous to investigate the performance of using only Detic+MT without incorporating the knowledge graph distillation technique.

5. Cropping the object regions and using CLIP for recognition could considerably slow down the training speed of the model. The paper should offer a detailed explanation regarding this aspect.

**Questions:**

Please refer to weaknesses.

**Limitations:**

No.

---

> ### Author Rebuttal · Authors · 2024-08-07
>
> **Response to Weaknesses-1**:
> Thanks for your comments. We would clarify that the motivation for using knowledge graphs is to explicitly and comprehensively extract CLIP knowledge for effectively de-noising pseudo labels generated by LVDs when adapting LVDs.
> On the other hand, directly utilizing CLIP to obtain pseudo-labels could also benefit unsupervised domain adaptation of LVDs, but it may be less effective. The reason lies in that knowledge graphs carry not only the information of each category but also inter-class relations, while pseudo-labels only carry the former information.
> As the table bleow, we conduct new experiments that adapt Detic with semi-supervised learning [Ref 1]  using CLIP-generated pseudo-labels. The experimental results show that our proposed KGD outperforms the semi-supervised learning using CLIP-generated pseudo-labels, validating the performance gain largely comes from our novel KGD designs instead of merely using pseudo-labels from CLIP.
>
> Method |AP50
> -|-
> Detic (Baseline, source-only) |46.5
> semi-supervised learning [Ref 1]  (using CLIP-generated pseudo-labels) |48.8
> KGD (Ours) |53.6
>
> [Ref 1] Dong-Hyun Lee. Pseudo-label: The simple and efficient semi-supervised learning method for deep neural networks. In Workshop on Challenges in Representation Learning, ICML, volume 3, page 2, 2013
>
> **Response to Weaknesses-4**:
> Thanks for your comment. We would clarify that all methods (including MT [53] and Ours) in all tables use Detic [81] as the baseline in the manuscript. In another word, the MT [53] in all tables actually denote the mentioned “Detic [81]+MT [53]”.
>
> **Response to Weaknesses-5**:
> Thanks for your comment. We would clarify that we studied the training and inference time of all the compared methods in the manuscript, and Table 11 in the manuscript shows the results on Cityscapes. It shows that incorporating CLIP into unsupervised domain adaptation introduces a few additional computation overhead in training time and almost does not affect the inference time. The reason lies in that we only crop a limited number of object regions (i.e., selected ones) and process them with CLIP model in a parallel manner during training, while the inference pipeline does not involve these procedures.

---

> ### Author Response · Authors · 2024-08-07
> **Response to Weaknesses-2&3**
>
> **Response to Weaknesses-2:**
>
> Thanks for your comments.
> As suggested, we introduce the Periodically Exchange Teacher-Student (PETS) [Ref 1] and Target Prediction Distribution Searching (TPDS) [Ref 2] as the comparison methods.
> As the table below shows, KGD consistently outperforms PETS [Ref 1] and TPDS [Ref 2].
>
> In the tables below, $\star$ signifies that the methods employ WordNet to retrieve category definitions given category names, and CLIP to predict classification pseudo labels for objects. We adopt AP50 in evaluations. The results of all methods are acquired with the same baseline (Detic[81]) as shown in the first row.
> We will include the new experiments into the updated paper later. Thank you for your suggestion!
>
>
> [Ref 1] Liu, Qipeng, et al. "Periodically exchange teacher-student for source-free object detection." Proceedings of the IEEE/CVF International Conference on Computer Vision. 2023.
>
> [Ref  2] Tang, Song, et al. "Source-free domain adaptation via target prediction distribution searching." International journal of computer vision 132.3 (2024): 654-672.
>
> Method |Cityscapes |Vistas |BDD100K-weather-rain|BDD100K-weather-snow|BDD100K-weather-overcast|BDD100K-weather-cloudy|BDD100K-weather-foggy |BDD100K-time-of-day--daytime |BDD100K-time-of-day--dawn&dusk|BDD100K-time-of-day--night
> -|-|-|-|-|-|-|-|-|-|-
> Detic (Baseline) |46.5 |35.0 |34.3 |33.5 |39.1 |42.0 |28.4 |39.2 |35.3 |28.5
> PETS |50.2 |35.8 |34.4 |33.9 |40.1 |43.0 |36.3 |39.7 |35.7 |27.8
> PETS$\star$ |50.8 |37.4 |35.9 |36.3 |41.0 |42.8 |36.7 |40.9 |37.2 |27.7
> TPDS |50.1 |36.0 |35.8 |35.2 |40.0 |42.1 |36.4 |40.4 |36.5 |28.5
> TPDS$\star$ |50.3 |37.1 |35.6 |35.9 |40.5 |43.4 |36.9 |41.3 |36.7 |28.9
> KGD (Ours) |53.6 |40.3 |37.3 |37.1 |44.6 |48.2 |38.0 |46.6 |41.0 |31.2
>
>
> Method |Common Objects: VOC|Common Objects: Objects365 |Intelligent Surveillance: MIO-TCD |Intelligent Surveillance:BAAI  |Intelligent Surveillance:VisDrone |Artistic Illustration:Clipart1k|Artistic Illustration:Watercolor2k|Artistic Illustration:Comic2k
> -|-|-|-|-|-|-|-|-
> Detic (Baseline) |83.9 |29.4 |20.6 |20.6 |19.0 |61.0 |58.9 |51.2
> PETS |85.9 |31.5 |20.6 |22.6 |18.2 |63.0 |60.2 |50.4
> PETS$\star$ |86.3 |32.1 |21.1 |23.2 |19.3 |63.6 |61.3 |50.6
> TPDS |85.5 |31.8 |20.2 |22.1 |18.8 |63.1 |60.0 |50.1
> TPDS$\star$ |85.6 |32.0 |21.1 |23.2 |19.2 |64.3 |61.4 |50.6
> KGD (Ours) |86.9 |34.4 |24.6 |24.3 |23.7 |69.1 |63.5 |55.6
>
> **Response to Weaknesses-3:**
> Thanks for your comment. As suggested, we conduct new experiments to compare our KGD with prior knowledge graph distillation methods[Ref 3, Ref 4] . The results in the tables below show that our KGD outperform [Ref 3, Ref 4] clearly, largely becuase the knowledge graphs in [Ref 3, Ref 4] are hand-crafted by domain experts while ours is built and learnt from CLIP. We will include the new experiments into the updated paper later.
>
>
> Method |Cityscapes |Vistas |BDD100K-weather-rain|BDD100K-weather-snow|BDD100K-weather-overcast|BDD100K-weather-cloudy|BDD100K-weather-foggy |BDD100K-time-of-day--daytime |BDD100K-time-of-day--dawn&dusk|BDD100K-time-of-day--night
> -|-|-|-|-|-|-|-|-|-|-
> Detic (Baseline) |46.5 |35.0 |34.3 |33.5 |39.1 |42.0 |28.4 |39.2 |35.3 |28.5
> KGE [Ref 5] |48.9 |36.0 |35.5 |34.4 |40.5 |41.2 |29.7 |40.1 |36.6 |29.0
> Context Matters [Ref 6] |49.4 |36.6 |36.3 |35.0 |41.7 |42.4 |30.2 |41.5 |37.2 |29.7
> KGD (Ours) |53.6 |40.3 |37.3 |37.1 |44.6 |48.2 |38.0 |46.6 |41.0 |31.2
>
>
> Method |Common Objects: VOC|Common Objects: Objects365 |Intelligent Surveillance: MIO-TCD |Intelligent Surveillance:BAAI  |Intelligent Surveillance:VisDrone |Artistic Illustration:Clipart1k|Artistic Illustration:Watercolor2k|Artistic Illustration:Comic2k
> -|-|-|-|-|-|-|-|-
> Detic (Baseline) |83.9 |29.4 |20.6 |20.6 |19.0 |61.0 |58.9 |51.2
> KGE [Ref 5] |85.4 |31.2 |20.3 |23.5 |19.4 |62.4 |58.1 |50.5
> Context Matters [Ref 6] |85.9 |31.7 |20.9 |23.3 |19.9 |62.9 |59.1 |52.3
> KGD (Ours) |86.9 |34.4 |24.6 |24.3 |23.7 |69.1 |63.5 |55.6
>
> [Ref 3] Christopher Lang, Alexander Braun, and Abhinav Valada. Contrastive object detection using knowledge graph embeddings. arXiv preprint arXiv:2112.11366, 2021
>
> [Ref 4] Aijia Yang, Sihao Lin, Chung-Hsing Yeh, Minglei Shu, Yi Yang, and Xiaojun Chang. Context matters: Distilling knowledge graph for enhanced object detection. IEEE Transactions on Multimedia, 2023, doi: 10.1109/TMM.2023.3266897.

---

> > ### Comment · Reviewer_Jwgx · 2024-08-13
> >
> > Thanks for the detailed response which has addressed most of my concerns. Nevertheless, the contribution is still believed to be not significant. I'd like to raise my rating to borderline accept.

---

### Official Review · Reviewer_TyHq · 2024-07-13

**Soundness:** 3
**Presentation:** 2
**Contribution:** 2
**Rating:** 6
**Confidence:** 4

**Summary:**

The authors propose a method for domain adaptation for large-vocabulary object detectors. To perform the adaptation, the authors first construct a Language and a Vision graph from the set of classes of the target dataset. The language graph is built with nodes as the description of the target class and the hyponym set of the class as retrieved from WordNet, and each description is encoded with the text encoder of CLIP to define the nodes’ features. This graph is exploited to adapt and improve the predictions on an image with a graph convolutional network (GCN). The Vision graph is initialized with nodes for all classes of the target dataset, with nodes represented by the text embedding of each class which are then adapted using the visual embedding centroid of each category. The similarity between the bounding boxes’ features and these adapted visual nodes’ representation is exploited to improve the class predictions. Experiments conducted on multiple datasets with multiple baseline comparisons show state-of-the-art performance of the proposed method. Ablation studies also show the contribution of each component of the method.

**Strengths:**

- The authors present a sound method for domain adaptation for large-vocabulary object detectors exploiting graphs built from the textual and visual modalities from the set of classes of the target dataset for adaptation
- The proposed method achieves state-of-the-art performance across a wide range of object detection datasets
- Detailed ablation studies show the contribution of each component

**Weaknesses:**

- The main weakness of this paper may be in the terminology used by the authors. For example, the authors talk about Knowledge Graphs for things that are not aligned with the common meaning of these terms, as the edges in the LKG and VKG graphs do not convey semantic relationships but are simply affinity edges based on the distance between the nodes' representation. On the contrary, the authors could name more explicitly the underlying methods and principles adopted in their approach, e.g. the step in Eq. (7) seems to perform a semi-supervised label propagation with a graph convolutional network, and Eq. (13) is performing embedding propagation. The term “encapsulation” may not be well adapted either, and it also hides the main visual adaptation steps in Eq. (11)  and (12). Overall, this hinders the readability of the paper and induces confusion.
- Some elements of the method should be detailed (see questions)
- Some experimental results could be presented more clearly (see questions)

**Questions:**

- Figure 2, what do the dashed reddish lines between LKG and VKG represent?
- In the WordNetRetrieve step, it is unclear if the hyponym set just contains the hyponym class names, their descriptions, or both. Similarly, is the initial class name used?
- p6 Eq. (12), what is the value of $\lambda$? How is it selected? How sensitive is the method to this value?
- p6, under Eq. (13), from the reference (54) it is not immediately clear what is the value of $\alpha$, how it selected, and how sensitive the method would be to this value. The authors should discuss this directly in the paper.
- Ablation studies tables (Table 3, 5, 6, 7) are somewhat redundant and hard to read due to the layout. The authors could invert rows and columns and combine all these tables into one to improve readability. Interestingly, it seems the most simple “LKG Extraction with category name” and “Static VKG Extraction” already show significant improvement over the baseline 46.5 -> 51.9, while adding all the other elements of the method brings less than one more point of performance. Do the authors have the results of using these two jointly? How far would that be from the whole KGD (Ours) results? Ablation results are only reported on the Cityscapes dataset, have the authors conducted similar analyses on the other datasets? If so, is the behavior consistent across datasets?

Typos etc:
- p6-l195: this sentence is unclear “the update-to-date object features to update it using manifold smoothing”
- p9-l306-311: use “smoothing” instead of “smooth”

**Limitations:**

Yes, in A.4.8.

---

> ### Author Rebuttal · Authors · 2024-08-07
>
> **Response to Weakness-1**: Thanks for your detailed comments. In our context, the edges in LKG and VKG represent affinity edges based on the distance between node representations. On the other hand, we model these distances as the semantic similarities measured by CLIP model, where the CLIP model largely captures rich semantic information among various categories and the resulted distances thus reflect meaningful semantic relationships among different nodes. We will clarify this distinction in the revised manuscript. The term “encapsulation” was to convey the process of integrating extracted knowledge graphs into the object detector. We acknowledge that this term may obscure the visual adaptation steps described in Eq. (11) and Eq. (12).
> We will provide detailed explanations to accurately reflect these adaptation in the revised manuscript.
>
> **Response to Weakness-2**: Thanks for your comments! We will check through our manuscript text carefully and improve the paper presentation later. Please refer to the responses to Question 1-2.
>
> **Response to Weakness-3**: Thank you for your comments! As suggested, we provide detailed explanations and conducted experiments for your concerns. Please refer to the responses to Question 3-5.
>
> **Response to Question-1**:
> The dashed reddish lines between LKG and VKG in Figure 2 represent the cross-modal edges that connect the nodes between vision and language modalities.
> The purpose of these edges is to enable the integration of both language and visual information, allowing the model to leverage complementary information from both modality.
> We will revise the caption of Figure 2 and include a more detailed explanation in the main text to ensure this is clear to the readers.
> Thanks for your comments!
>
> **Response to Question-2**:
> In this paper, the hyponym set contains both the hyponym class names and their descriptions.
> Specifically, each hyponym in the hyponym set is the concatenation of the class name and its descriptions, which contains both the class names and their description.
> In the same way, the initial class name is used by concatenating the initial class name and its description to formulate the definition of a certain category as the following:
>
> [*class name*]+[':']+[*category description*]
>
>
> **Response to Question-3**:
> In the Eq. (12), the nodes of VKG are preliminarily updated with a pre-defined $\lambda$.
> $\lambda$ is set as 0.9999.
> We study $\lambda$ by changing it from 0.99 to 0.999999.
> The table below reports the experiments over the Cityscapes dataset.
> It shows that both an excessivel small $\lambda$ or excessively large $\lambda$ lead to performance degradation,
> largely because an excessively small $\lambda$ (i.e., 0.99) introduces more noise and fluctuation, while an excessively large $\lambda$ (i.e., 0.999999) results in a sluggish response to the latest data changes, failing to update VKG nodes promptly.
> However, an appropriate value (0.9999) of $\lambda$ can suppress noise and data fluctuation while promptly updating VKG nodes to timely respond to the latest data distribution shift.
>
> $\lambda$  |0.99      |0.999      |0.9999      |0.99999      |0.999999
> -------- | ------------- | ------------- | ------------- | ------------- | -------------
> AP50  |49.9          |51.5           |53.6          |52.3           |51.8
>
> **Response to Question-4**:
> Eq. (13), incorporate the downstream visual graph knowledge into VKG with a pre-defined $\alpha$.
> $\alpha$ is set as 0.001.
> We study $\alpha$ by changing it from 0.001 to 1.0.
> The table below reports the experiments over the Cityscapes dataset.
> It shows that both an too small $\alpha$ or too large $\alpha$ lead to performance degradation,
> largely because a too small $\alpha$ may cause the model to fail to effectively utilize the information from neighboring nodes, thus not fully capturing the structure of the graph and the relationships between nodes, while a too large $\alpha$ may cause noise to propagate through the graph, making the node updates more susceptible to outliers or noisy data.
>
> $\alpha$     |0.0001      |0.001      |0.01      |0.1      |1
> -------- | ------------- | ------------- | ------------- | ------------- | -------------
> AP50    |50.9 |51.0 |53.6   |49.9 |49.2
>
>
>
>
>
> **Response to Question-6**:
> Thank you for your detailed comments! We will carefully check and improve the paper presentation in the revised manuscript.
>
> **Response to Question-7**:
>
> Thank you for pointing these issues and we will revise the paper accordingly such that all typos are corrected. We will make the following revisions:
>
> ''Dynamic VKG Extraction without Smooth''-->''Dynamic VKG Extraction without Smoothing''.
>
> ''but without smooth''-->''but without smoothing''.

---

> ### Author Response · Authors · 2024-08-07
> **Response to Question-5**
>
> **Response to Question-5**:
>
> Thanks for your detailed comments! We will check through our manuscript carefully and improve the table layout in the revised manuscript. As suggested, we conduct experiments and report the results of using ''LKG Extraction with category name'' and ''Static VKG Extraction'' jointly in the following table.
> As a comparison, our proposed KGD shows clear improvements as the language and vision information extracted along the training process dynamically stabilizes and improves the model adaptation, validating the performance gain largely comes from our novel KGD designs instead of jointly using ''LKG Extraction with category name'' and ``Static VKG Extraction''.
>
> Method |Detic (Source only)||||KGD
> -|-|-|-|-|-
> LKG Extraction with category names ||$\checkmark$||$\checkmark$|
> Static VKG Extraction |||$\checkmark$|$\checkmark$|
> LKG Extraction with WordNet Hierarchy |||||$\checkmark$
> Dynamic VKG Extraction |||||$\checkmark$
> AP50 |46.5|51.9|51.9|52.4|**53.6**
>
> As suggested, we conduct additional ablation study on 3 object detection datasets that span different downstream domains including the object detection for intelligent surveillance (BAAI), common objects (VOC), and artistic illustration (Clipart1k).
> As the table below shows, the behavior consistent across datasets that span different downstream domains.
> Later, We will conduct ablation study on all the 11 object detection datasets and include the results into the revised manuscript. Thank you for your suggestion!
>
> Method | Language Knowledge Graph Distillation | Vision Knowledge Graph Distillation |AP50|AP50|AP50|AP50
> -|-|-|-|-|-|-
> Dataset  | | |Cityscapes|BAAI|VOC|Clipart1k
> Detic (Baseline)|||46.5|20.6|83.9|61.0
> ||$\checkmark $||52.8|22.2|86.1|66.5
> |||$\checkmark$|52.7|22.4|86.2|67.1
> KGD (Ours)|$\checkmark $|$\checkmark $|53.6|24.3| 86.9| 69.1

---

> ### Comment · Reviewer_TyHq · 2024-08-11
>
> I have read all the reviewers' comments and the authors' answers, I believe my initial rating of `6: Weak Accept` is a fair assessment for this contribution and thus maintain it.

---

### Decision · Program_Chairs · 2024-09-25

**Decision:**

Accept (poster)

**Comment:**

This work exploits the implicit knowledge graphs (KG) in CLIP to effectively adapt LVDs to various downstream domains. During the rebuttal period, authors have addressed the concerns on computation overhead and insufficient and unfair experimental comparison raised by a few reviewers and all reviewers rate positively. Authors should include all results in the rebuttal in the revision. After careful consideration, The AC is inclined to accept this paper.